# Manipulating Zika virus RNA tertiary structure for developing tissue-specific attenuated vaccines

Xiang Chen[1,4], Meng-Li Cheng[2,4], Xing-Yao Huang[1,4], Meng-Xu Sun[1,4], Rui-Ting Li [ID][1,4], Mei Wu[1], Yu-Yan Li[1], Qian Xu[1], Meng-Yue Guan[1], Hui Zhao[1] & Cheng-Feng Qin [ID] [1,3✉]

## Abstract

**Traditional live attenuated vaccines (LAVs) are typically developed through serial passaging or genetic engineering to introduce specific mutations or deletions. While viral RNA secondary or tertiary structures have been well-documented for their multiple functions, including binding with specific host proteins, their potential for LAV design remains largely unexplored. Herein, using Zika virus (ZIKV) as a model, we demonstrate that targeted disruption of the primary sequence or tertiary structure of a specific viral RNA element responsible for Musashi-1 (MSI1) binding leads to a tissue-specific attenuation phenotype in multiple animal models. The engineered MSI1-binding-deficient ZIKV mutants (MBD) maintained full competence in MSI1-deficient tissues but were significantly restricted in ZIKV-vulnerable tissues (brain, testis, eye and placenta) and exhibited substantially reduced vertical transmission in mice. Importantly, a single immunization with MBD ZIKV induced robust immune responses and conferred protection against ZIKV challenge in both mice and non-human primates. Thus, our study demonstrates that manipulating viral RNA structures that interact with host proteins represents a powerful platform for developing the next generation of LAVs against emerging viruses.**

**Keywords** RNA Structure; Zika Virus; Musashi-1; Live Attenuated Vaccine
**Subject Categories** Immunology; Microbiology, Virology & Host Pathogen Interaction

## Introduction

Live attenuated vaccines (LAVs) represent one of the most effective methods of immunization because they mimic the natural life cycle of wild-type (WT) viruses and are capable of activating all components of the immune systems at a low production cost. An ideal LAV should be sufficiently attenuated for safety while retaining robust immunogenicity for efficacy and being producible efficiently in manufacturing-suitable tissue culture platforms. Generally, LAV candidates are obtained by serially passaging in susceptible cells/animals or by genetic modification using reverse genetics technology. Over the last two decades, several rational and innovative approaches to viral attenuation have been developed, such as controlling replication fidelity (Vignuzzi et al, 2008), altering codon pair bias (Coleman et al, 2008), inserting microRNA binding sites (Barnes et al, 2008), incorporating mutations sensitive to interferon or zinc finger antiviral protein (Du et al, 2018; Goncalves-Carneiro et al, 2022), and proteolytic-targeting chimeric (PROTAC) (Si et al, 2022). These advanced platforms target either the viral proteins or the primary sequences of viral RNA. With advances in structural and RNA biology technologies, the higher-order RNA structures within viral genomes have been identified with specific functions (Akiyama et al, 2016; Bhatt et al, 2021; Bonilla et al, 2021; Brown et al, 2020; Chapman et al, 2014; Keane et al, 2015; Teng et al, 2022). However, whether viral RNA structures can be targeted for vaccine or antiviral development remains less investigated.

Zika virus (ZIKV) is a single-stranded, positive-sense RNA virus in the *Flaviviridae* family. ZIKV infection has been linked to microcephaly and other congenital defects in infants, and Guillian-Barré syndrome in adults. Despite several promising ZIKV LAV candidates have been described, including chimeric vaccines and genetically modified vaccines (Baldwin et al, 2021; Chin et al, 2021; Giel-Moloney et al, 2018; Kum et al, 2018; Li et al, 2018a; Li et al, 2018b; Shan et al, 2017; Touret et al, 2018; Xie et al, 2018; Zhong et al, 2022), none of them has been licensed yet. Distinct from other flavivirus members, ZIKV exhibits specific tissue tropism. While ZIKV replication in visceral organs (e.g., heart, liver, intestine, kidney, and lung) does not cause obvious tissue damage (Aliota et al, 2016; Hirsch et al, 2017), infection of the brain, eye, testis and placenta can lead to severe diseases including microcephaly, ocular abnormalities and testicular damage in mouse models (Brasil et al, 2016; Cao-Lormeau et al, 2016; Govero et al, 2016; Li et al, 2016a; Li et al, 2016b; Li et al, 2021; Ma et al, 2016; Salinas et al, 2017; Simonin et al, 2019; Tang et al, 2016; Uraki et al, 2017). Thus, abolition of specific tissue tropism of ZIKV represents a promising attenuation strategy.

Recently, we identified a unique RNA element, termed MBS, within the 3′untranslated region (UTR) of ZIKV genome. The

[1]State Key Laboratory of Pathogen and Biosecurity, Academy of Military Medical Sciences, 100071 Beijing, China. [2]Experimental platform management office, Beijing Key Laboratory of Drug-Resistant Tuberculosis, Beijing Chest Hospital, Capital Medical University, Beijing Tuberculosis and Thoracic Tumor Institute, 101149 Beijing, China. [3]Research Unit of Discovery and Tracing of Natural Focus Diseases, Chinese Academy of Medical Sciences, 100071 Beijing, China. [4]These authors contributed equally: Xiang Chen, Meng-Li Cheng, Xing-Yao Huang, Meng-Xu Sun, Rui-Ting Li. ✉E-mail: qincf@bmi.ac.cn

primary sequence and tertiary structure of this element are critical for its specifically binding with the host protein Musashi-1 (MSI1) (Chen et al, 2023). MSI1, an evolutionarily conserved RNA-binding protein, is highly enriched in undifferentiated neural stem cells (NSCs) but gradually declined following development (Sakakibara et al, 1996). MSI1 has been linked to ZIKV-induced microcephaly (Chavali et al, 2017). Further characterization revealed that recombinant ZIKV mutants deficient in MSI1 binding were restricted in MSI1-expressing cells (Chen et al, 2023). In the present study, we sought to leverage the unique sequence and tertiary structure of the MBS RNA element to rationally design and develop a novel LAV by using reverse genetics platform. Our results clearly demonstrated that disruption of the primary sequence or tertiary structure of MBS leads to tissue-specific attenuation and induces protective immunity against ZIKV challenge in both mice and non-human primates. These findings expand the arsenal to develop LAVs by targeting viral RNA structures against emerging viruses.

## Results

### MBD ZIKV exhibits tissue-specific attenuation

To leverage the unique tissue distribution pattern of MSI1 in ZIKV-susceptible hosts, we first analyzed mRNA expression levels in humans using Human Protein Atlas (HPA). As shown in Fig. 1A, the retina, testis, and cerebral cortex exhibit the highest MSI1 mRNA expression levels in human. Further validation in 4-week-old A129 mice by quantitative RT-PCR (RT-qPCR) and Western blotting revealed that MSI1 is specifically enriched in the brain, testis, and eye (Fig. 1B,C). Immunocytochemistry results further confirmed high levels of MSI1 expression in these ZIKV-vulnerable organs (Fig. EV1).

Herein, specific mutants targeting MSI1 binding site (MBS) were introduced into the infectious clone of ZIKV to engineer two MSI1-binding-deficient variants: MBD1 and MBD2 (Fig. 1D). For MBD1, the primary sequence of MBS1 was altered from "AUAG" to "AAAG"; While MBD2 modified the tertiary structure of MBS2 from an AGAA tetraloop to a GAAA tetraloop (Chen et al, 2023). The infection and replication of the two mutants in cells associated to ZIKV-specific tissue tropism were then assessed. As expected, in a human retina-derived cell line Y79, which has a high expression level of MSI1 (Fig. EV2A), MBD1 and MBD2 showed significantly restricted replication compared to WT ZIKV (Fig. EV2B). This result is consistent with our previous findings in human brain cell lines that express MSI1 (Chen et al, 2023). In contrast, MBD1 and MBD2 exhibited similar replication efficiency to WT ZIKV in Vero cells (Fig. EV3A,B), which are deficient in MSI1. To rule out the possible contribution of host interferon (IFN) response, we assessed viral replication kinetics and downstream IFN induction in human A549 cells (MSI1-deficient). The results demonstrated WT and MBD mutants exhibited comparable replication efficiency (Fig. EV3C) and comparable induction of IFN-β and IFIT1 mRNA levels (Fig. EV3D,E). Additionally, to determine whether the introduced mutations affect the production of subgenomic flavivirus RNA (sfRNA), we profiled the sfRNA expression pattern of MBD mutants versus WT virus by Northern blotting in human SH-SY5Y (MSI1-proficient) and A549 (MSI1-deficient) cells

(Fig. EV3F). The results showed that while the sfRNA patterns differed between SH-5Y5Y and A549 cells, there was no difference in expression between WT and the MBD mutants (Fig. EV3G).

To further investigate the in vivo replication kinetics and tissue distribution of the two MBD mutants, groups of adult A129 mice were inoculated with WT ZIKV, MBD1 or MBD2. At 5 days post-infection (dpi), mice infected with MBD1 or MBD2 showed significantly lower viral RNA loads in the brain, eyes, and testes compared to those infected with WT ZIKV (Appendix Fig. S1A–C), while in serum and other organs, the viral RNA loads of MBD1 and MBD2 were comparable to those of WT ZIKV (Appendix Fig. S1D–J). Moreover, the viral titers determined by plaque assays in the brain, eyes, testes and serum further confirmed these findings (Fig. 1E–H). These results suggest MBD ZIKV exhibits specific attenuation in tissues with high MSI1 expression, which are vulnerable to ZIKV infection.

Since ZIKV infection can damage testes in mice (Govero et al, 2016; Ma et al, 2016; Uraki et al, 2017), we further evaluated the infectivity and replication of MBD1 and MBD2 in the mouse testes. In the testicular tissues infected with WT ZIKV, substantial ZIKV E-positive cells were observed in adjacent clusters, with signals distributed throughout the seminiferous epithelium. In contrast, only a small number of positive cells were detected in a scattered distribution in the testicular tissues infected with MBD1 or MBD2 (Fig. 1I). Histopathological analysis further corroborated that WT ZIKV-infected mice exhibited marked disruption of normal testicular architecture. In contrast, MBD1 or MBD2-infected mice maintained normal testicular phenotypes (Fig. 1J). These results demonstrate MBD ZIKV is highly restricted in the testes of mice.

### MBD ZIKV is highly attenuated in mouse brains and human brain organoids

Given that MSI1 is highly expressed in human and mouse neural progenitor cells (Chavali et al, 2017; Kaneko et al, 2000; Sakakibara et al, 1996), we next investigated the neurovirulence of MBD1 and MBD2 using the well-established 1-day-old neonatal mouse model (Yuan et al, 2017). Upon intracerebral injection with $1 \times 10^3$ PFU of virus, WT ZIKV led to 58.3% mortality in neonatal mice. In contrast, all mice infected with MBD1 or MBD2 survived (Fig. 2A). Furthermore, mice infected with either MBD1 or MBD2 did not exhibit the microcephaly phenotype that was observed in WT ZIKV-infected mice (Fig. 2B). Consistent with the survival outcome, viral titers in brains of MBD1 or MBD2-infected mice were significantly lower than WT ZIKV-infected mice (Fig. 2C). IFA showed that the viral signal intensity in the cortex and hippocampus of MBD1 or MBD2-infected mice was substantially lower than that in WT ZIKV-infected counterparts (Fig. 2D,E). These results demonstrate that MBD1 and MBD2 are highly attenuated in mouse neurovirulence.

Given that MBD1 and MBD2 showed similar attenuation phenotypes in the aforementioned experiments, we selected the MBD2 mutant, which contains a disrupted tertiary structure of MBS2, for subsequent experiments to highlight the structural novelty and avoid redundancy. To compare the host immune response at transcriptomic level, RNA-seq was performed with neonatal mouse brains infected with WT or MBD2 ZIKV. In MBD2-infected animals, 189 genes were upregulated and 606 genes were downregulated compared to WT-infected animals (Appendix

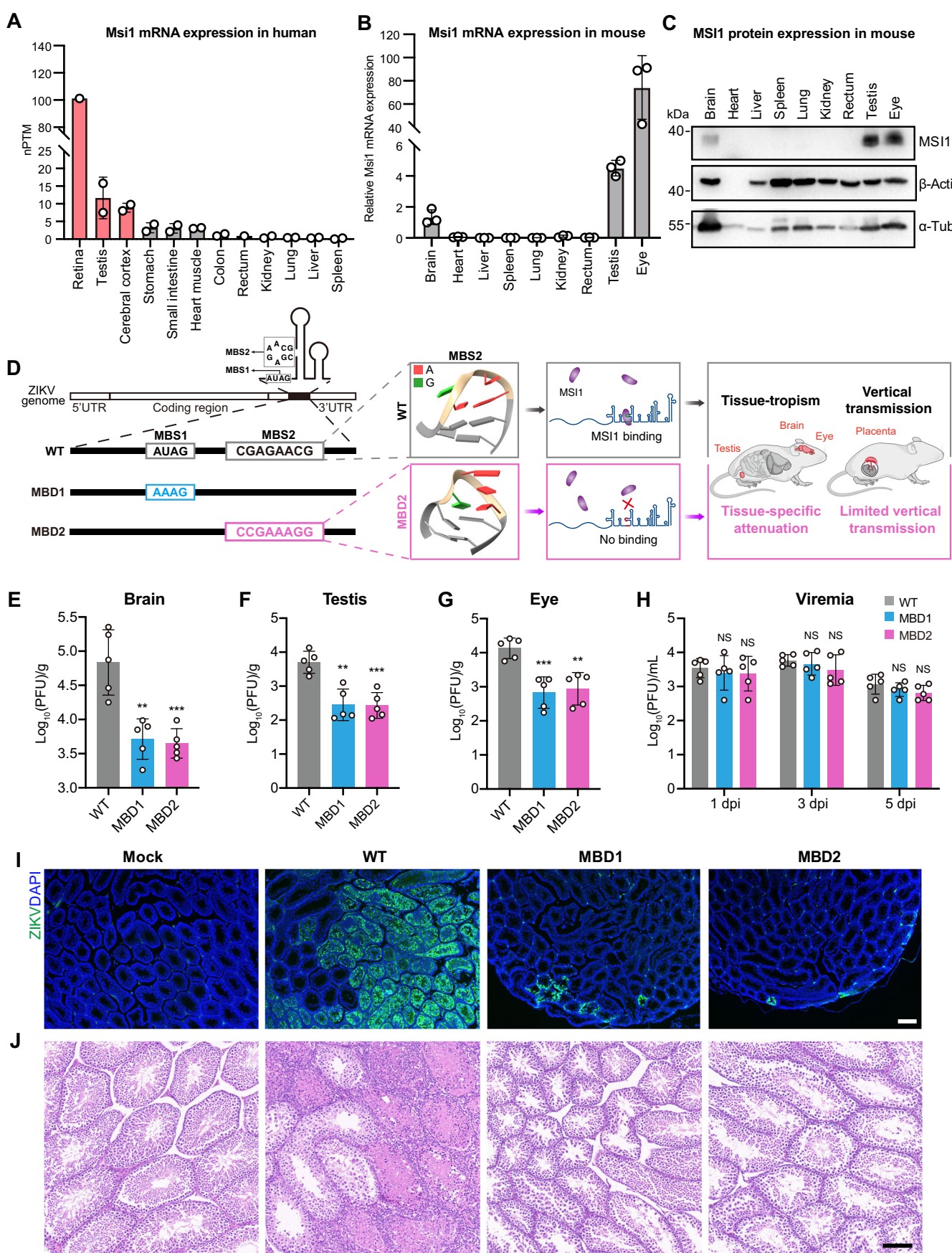

**Figure 1. MBD ZIKV is specifically attenuated in tissues vulnerable to ZIKV.**

(A) Msi1 mRNA expression levels (nTPM, normalized transcripts per million) in different human tissues. Data were obtained from the HPA and GTEx datasets (https://www.proteinatlas.org/ENSG00000135097-MSI1/tissue). For Retina ($n = 1$), the value represents nTPM derived from the GTEx dataset. For Rectum ($n = 1$), the value represents nTPM derived from the HPA dataset. For other tissues ($n = 2$), values represent nTPM derived from both HPA and GTEx datasets. Error bars represent mean ± SD. (B) Relative Msi1 mRNA levels in different tissues of 4-week-old A129 mice determined by RT-qPCR. Data are the mean ± SD. Two-sided Student's $t$ test. $n = 3$. $n$ represents biological replicates. (C) MSI1 protein levels in different organs of mouse determined by Western blotting. The Actin and Tubulin act as loading controls. (D) Schematic illustration of the generation of the MBD ZIKV that specifically attenuated in tissues vulnerable to ZIKV. The secondary structure of the stem-loop containing MBS1 and MBS2 is shown, where the MBS1 and MBS2 motifs are presented by letters. The MBS2 tertiary structures of WT and MBD2 are represented as ribbon diagrams. MBS, MSI1 binding site; MBD, MSI1 binding deficient. (E–H) Viral titers in organs of infected A129 mice. Four-week-old A129 mice were infected with $1 \times 10^4$ PFU of WT or MBD ZIKV. Organs and blood samples from infected mice were collected and homogenized on 5 dpi. The viral titers were quantified by plaque forming assay. Data are the mean ± SD. $n = 5$. $n$ represents biological replicates. Two-sided Student's $t$ test. **$P < 0.01$, ***$P < 0.001$, NS, not significant ((E) MBD1 $P = 0.0021$, MBD2 $P = 0.0010$; (F) MBD1 $P = 0.0011$, MBD2 $P = 0.0005$; (G) MBD1 $P = 0.0008$, MBD2 $P = 0.0015$). (I) 4-week-old A129 mice were subcutaneously infected with $1 \times 10^4$ PFU of WT ZIKV, MBD1 or MBD2. Testes were collected on 14 dpi. Testis sections were immunostained with an anti-ZIKV-E antibody. Scale bar, 200 μm. (J) Histopathological examination of testis sections from infected animals. Scale bar, 100 μm. Source data are available online for this figure.

Fig. S2A). The top five downregulated Gene Ontology (GO) terms in MBD2-infected brains were primarily associated with positive regulation of immune response, inflammatory response, leukocyte activation, regulation of cell killing, and response to interferon-β (Appendix Fig. S2B). Heatmap analysis revealed that, compared to the Mock group, host response related to inflammatory and IFN-β (GO: 0006954 and GO:0035456) were commonly upregulated across the WT and MBD2 groups. However, the magnitude of upregulation was obviously attenuated in the MBD2 group relative to the WT group (Appendix Fig. S2C,D). As expected, brain development-related genes, including Padi4 (Bazan et al, 2023), Msx3 (Otsuki and Brand, 2019), Zfp558 (Johansson et al, 2022), Ltc4s (Mayatepek et al, 1999), H4c2 (Tessadori et al, 2022), Dct (Jiao et al, 2006), and Mid1 (Frank et al, 2024), were significantly downregulated in the WT infection group relative to the Mock group, while the expression of these genes were comparable between the MBD2 group and Mock group (Appendix Fig. S2E). These transcriptomic data further support the attenuation phenotype in mice.

To further expand the attenuation phenotype of MBD ZIKV in human tissues, human cerebral organoids derived from induced pluripotent stem cell (iPSC) were used as previously described (Lancaster et al, 2013). As shown in Fig. 3A,B, the organoids infected with WT ZIKV obviously shrank and lost their structure by day 5. In contrast, the organoids infected with MBD2 were similar in size and morphology to uninfected organoids. Moreover, MBD2 showed markedly reduced replication compared to WT ZIKV, as evidenced by lower levels of viral RNA (Fig. 3C) and envelope (E) protein (Fig. 3D). Collectively, these results demonstrate that MBD ZIKV is highly attenuated in human brain organoids.

## MBD ZIKV exhibits highly restricted infection of fetal brains during vertical transmission

ZIKV can be vertically transmitted from the maternal placenta to fetus (Miner et al, 2016), but the expression pattern of MSI1 in placenta has not been previously studied. To address this gap, we examined MSI1 expression in mouse placenta. Placentas ranging from embryonic day 8.5 (E8.5) to E12.5 were immunostained with an MSI1 antibody, revealing MSI1 expression throughout this developmental window (Figs. 4A and EV4). Notably, MSI1 was predominantly localized in trophoblast giant cells (Figs. 4A and EV4), which are positioned at the implantation site and

interface directly with the maternal decidua. Furthermore, MSI1 protein levels gradually decreased as placental development progressed (Figs. 4A and EV4). These results were further confirmed by Western blot analysis (Fig. 4B). The expression profile of MSI1 in placenta implies that MSI1 may play a role in the vertical transmission of ZIKV.

To evaluate the impact of MSI-binding deficiency on ZIKV vertical transmission, we assessed the maternal-to-fetal transmission capability of MBD2 using a well-established murine pregnancy model (Miner et al, 2016; Shan et al, 2020). Female CD-1 mice were mated and monitored for vaginal plug (defined as E0.5), followed by intraperitoneally administered anti-IFNAR1 antibodies at E5.5 to allow ZIKV infection (Fig. 4C). At E6.5, the pregnant mice were subcutaneously (s.c.) infected with $1 \times 10^5$ PFU of WT ZIKV or MBD2 virus. Animals were sacrificed at E13.5, and viral RNA loads were quantified in maternal and fetal organs (Fig. 4C). Consistent with the results in A129 mice (Appendix Fig. S1), no significant difference in viral RNA loads was observed in the maternal serum or spleen between WT ZIKV and MBD2 infections (Fig. 4D,E). However, MBD2-infected mice exhibited significantly lower viral RNA loads in maternal brains compared to those infected with WT ZIKV (Fig. 4F). Moreover, viral RNA loads in the placentas (Fig. 4G) and fetal heads (Fig. 4H) of MBD2 infections were significant lower than WT ZIKV infections. More importantly, about 40% of the fetal heads in MBD2 infections were negative for viral RNA, suggesting MBD2 is highly attenuated in vertical transmission. Corroborating with the in vivo results, experiments using the human placenta choriocarcinoma cell line Bewo which expresses MSI1 (Fig. EV5A), showed MBD2 replicates at a significantly slower rate than WT ZIKV (Fig. EV5B). These results clearly demonstrate highly restricted infection of neural tissues during vertical transmission of MBD ZIKV.

## MBD ZIKV elicits robust immune responses in mice

To evaluate the potential of MBD ZIKV as a LAV, groups of 4-week-old A129 mice were immunized with a single dose of WT ZIKV, and the two MBD mutants (Fig. 5A). Both MBD1 and MBD2 were analyzed in the vaccine immunization study to validate that structure-targeted attenuation (MBD2) confers protection comparable to conventional sequence disruption (MBD1). All mice received MBD1 and MBD2 viruses maintained body weight gains comparable to the PBS group, whereas those infected with WT virus exhibited significant weight loss (Fig. 5B). On day 28 post-

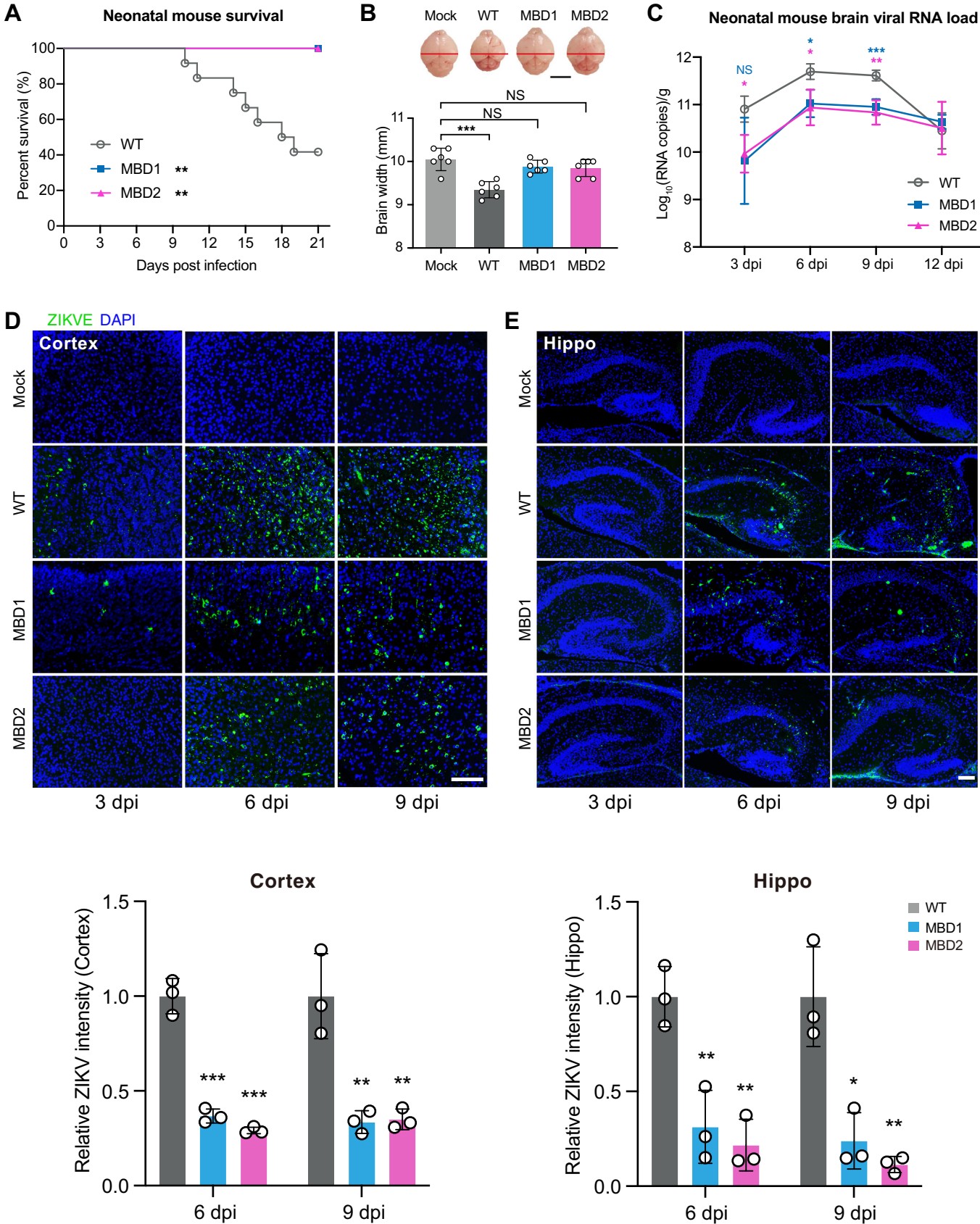

**Figure 2. MBD ZIKV exhibits highly attenuated neurovirulence in neonatal mice.**

(A) Survival curves of BALB/c neonatal mice intracerebrally injected with WT ZIKV ($n = 12$), MBD1 ($n = 12$), and MBD2 ($n = 11$) viruses at a dose of 1000 PFU. Analysis of survival was performed using log-rank test. **$P < 0.01$ (MBD1 $P = 0.0019$, MBD2 $P = 0.0029$). (B) Representative images of mouse brains from infected animals on day 9 post infection (Left panel). Brain width from infected mice was quantified (Right panel). Data are the mean ± SD. $n = 6$. $n$ represents biological replicates. Two-sided Student's $t$ test. ***$P < 0.001$ ($P = 0.0003$). Scale bar, 5 mm. (C) RNA from the ZIKV-infected whole brains was extracted and the viral RNA copies were determined by RT-qPCR. Data are the mean ± SD. $n = 3$. $n$ represents biological replicates. Two-way ANOVA. *$P < 0.05$, **$P < 0.01$, ***$P < 0.001$ (3 dpi: MBD2 $P = 0.0126$; 6 dpi: MBD1 $P = 0.0136$, MBD2 $P = 0.0295$; 9 dpi: MBD1 $P = 0.0005$, MBD2 $P = 0.0046$). (D, E) ZIKV-infected cortex (D) and hippocampus (hippo) (E) sections at indicated dpi were stained with anti-ZIKV-E antibody and quantification of relative intensity of ZIKV staining was analyzed. Data are the mean ± SD. Two-sided Student's $t$ test. (cortex, $n = 3$; hippo, $n = 3$). $n$ represents biological replicates. *$P < 0.05$, **$P < 0.01$, ***$P < 0.001$ ((D), 6 dpi: MBD1 $P = 0.0004$, MBD2 $P = 0.0002$; 9 dpi MBD1 $P = 0.0077$, MBD2 $P = 0.0082$; (E), 6 dpi: MBD1 $P = 0.0088$, MBD2 $P = 0.0029$; 9 dpi MBD1 $P = 0.0121$, MBD2 $P = 0.0045$). Scale bar, 100 μm. Source data are available online for this figure.

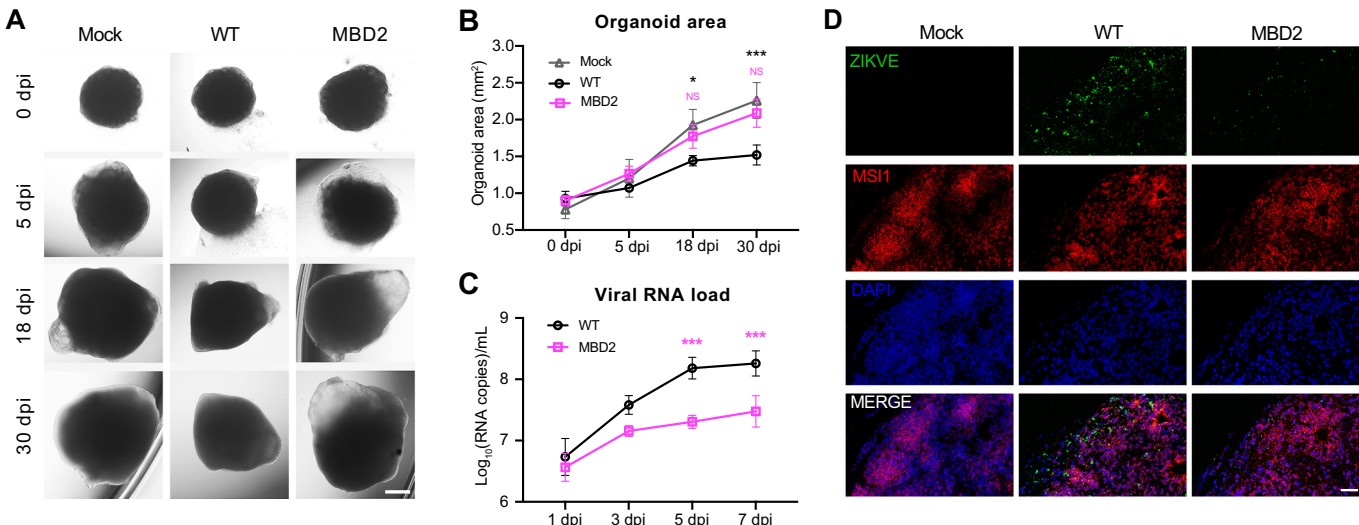

**Figure 3. MBD ZIKV is attenuated in human cerebral organoids.**

(A) Human cerebral organoids (hCOs) were infected with WT ZIKV or MBD2, and the Morphology of hCOs was monitored on indicated dpi. Scale bars, 500 μm. (B) Organoid area was measured on indicated time after infection. Data are the mean ± SD. $n = 3$. $n$ represents biological replicates. Two-way ANOVA, *$P < 0.05$, ****$P < 0.0001$ (18 dpi $P = 0.0101$, 30 dpi $P = 0.0002$). (C) Viral RNA loads in culture supernatants from hCOs infected with WT or MBD2 ZIKV in (A). Data are the mean ± SD. $n = 3$. $n$ represents biological replicates. Two-way ANOVA, ***$P < 0.001$ (5 dpi $P = 0.0003$, 7 dpi $P = 0.0008$). (D) The expression of ZIKV-E protein on day 7 after infection from (A) was detected by immunostaining. Scale bar, 40 μm. Source data are available online for this figure.

immunization, total IgG and neutralizing antibody titers against ZIKV in sera from immunized mice were measured using ELISA and PRNT assays, respectively (Fig. 5C,D). Both MBD1 and MBD2 induced high levels of ZIKV-specific IgG and neutralizing antibodies, comparable to those induced by WT infection. Following challenge with an epidemic ZIKV strain VEN/2016 on day 28 after immunization, all WT or MBD-immunized mice survived without detectable viremia or weight loss; In contrast, the PBS-immunized group exhibited significant viremia and weight loss and eventually succumbed (Fig. 5E–G). Overall, these results show that a single immunization of MBD ZIKV confers robust protection against ZIKV challenge in mice.

To assess T-cell responses in immunized mice, we performed enzyme-linked immunosorbent spot (ELISPOT) assay to detect the number of ZIKV-specific IFN-γ-secreting cells in MBD1 or MBD2-immunized mice on day 7 post immunization (Fig. 5H). The splenocytes were isolated from the spleen of immunized mice and stimulated with the ZIKV E protein. Much higher number of IFN-γ-secreting cells were observed in the MBD1 or MBD2-immunized mice compared to PBS-immunized controls in response to ZIKV E

stimulation (Fig. 5H). These data demonstrate that MBD ZIKV induces strong T-cell responses in mice after a single-dose inoculation.

Then we tested the genetic stability of MBD1 and MBD2. The viral stocks were subjected to 10 rounds of blind passages in Vero cells, which are approved cell lines for vaccine production. Sequencing of the 10th-passage (P10) viruses demonstrated that the MBD mutations remained intact after 10 rounds of passage (Appendix Fig. S3). To further test the genetic stability of these MBD mutants in vivo, we passaged the MBD1 and MBD2 viruses for 4 rounds in brains of 1-day-old neonatal mice, where MSI1 is highly expressed. The results of sequencing confirmed that the MBD mutations maintained after four rounds of passages in mice (Appendix Fig. S3).

## Immunogenicity and protection efficacy of MBD ZIKV in non-human primates

Finally, we further determine the immunogenicity and protection efficiency of MBD ZIKV in non-human primates. First, cynomolgus monkeys were prescreened as negative for flaviviruses (ZIKV, DENV,

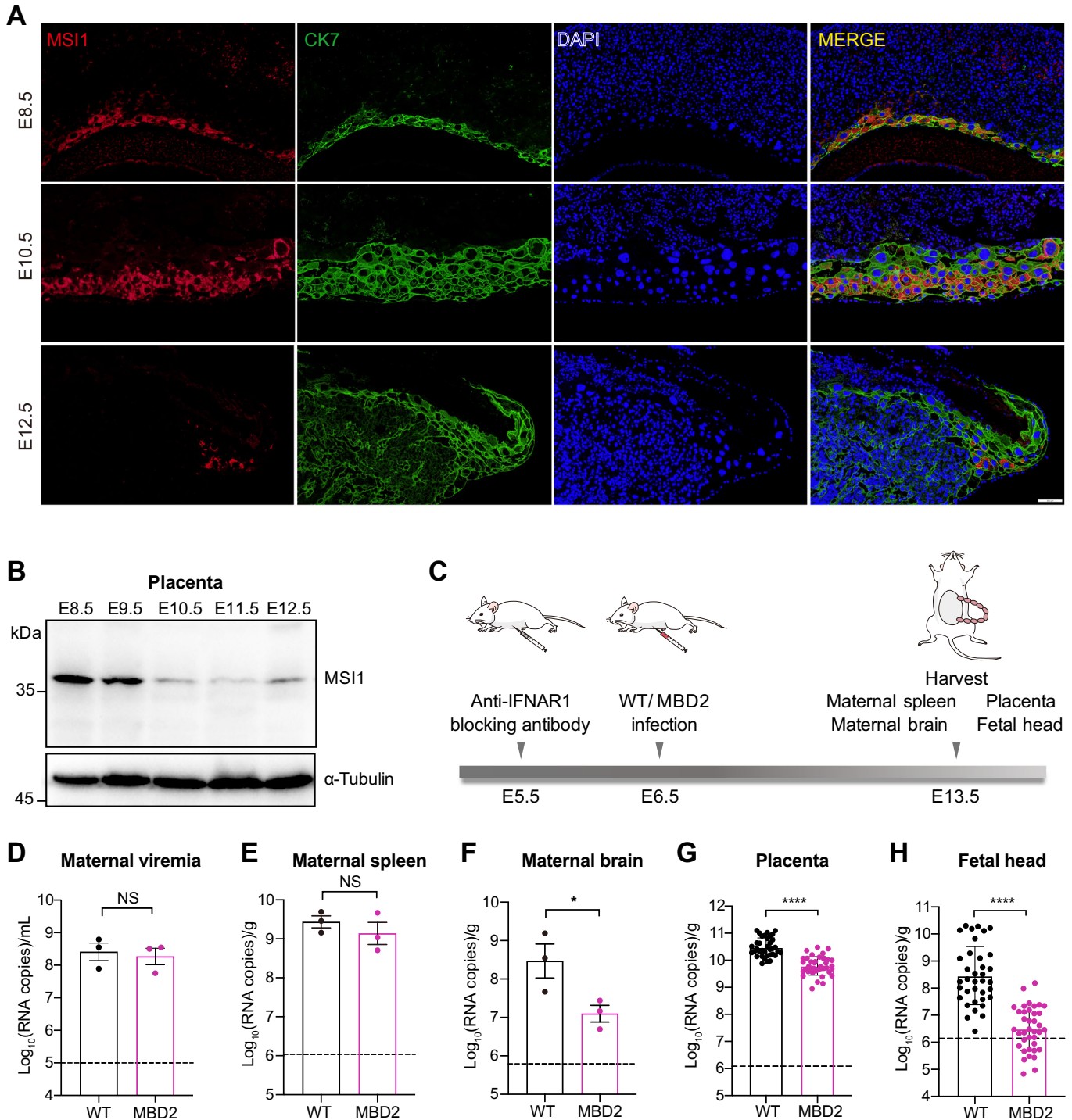

**Figure 4. MBD ZIKV exhibits highly restricted infection of fetal brains during vertical transmission.**

(A) Immunostaining of placenta sections from embryonic stage 8.5 (E8.5)-E12.5 mice with anti-MSI1 and anti-CK7 (trophoblast cell marker) antibodies. Scale bar, 100 µm. (B) Placentas from E8.5-E12.5 mice were lysed and MSI1 protein was detected by Western blotting. Tubulin acts as a loading control. (C) Experimental scheme. At E5.5, CD-1 mice were intraperitoneally administered with 2 mg of anti-IFNAR1 antibody. At E6.5, the mice were infected with $1 \times 10^5$ PFU of WT ZIKV or MBD2 via the s.c. route. At E13.5, maternal and fetal tissues were harvested and quantified for viral RNA loads using RT-qPCR. (D–H) Viral RNA loads are presented for maternal serum (D) ($n = 3$), spleen (E) ($n = 3$), maternal brain (F) ($n = 3$), placenta (G) (WT: $n = 36$; MBD2: $n = 38$), and fetal head (H) (WT: $n = 36$; MBD2: $n = 38$). $n$ represents biological replicates. The dashed line indicates the limit of detection (LOD) of the assay. Data are the mean ± SD. Two-sided Student's $t$ test. *$P < 0.05$, ****$P < 0.0001$ ((F) $P = 0.0497$, (G) $P = 1.25e-012$, (H) $P = 2.91e-013$), NS, not significant. Source data are available online for this figure.

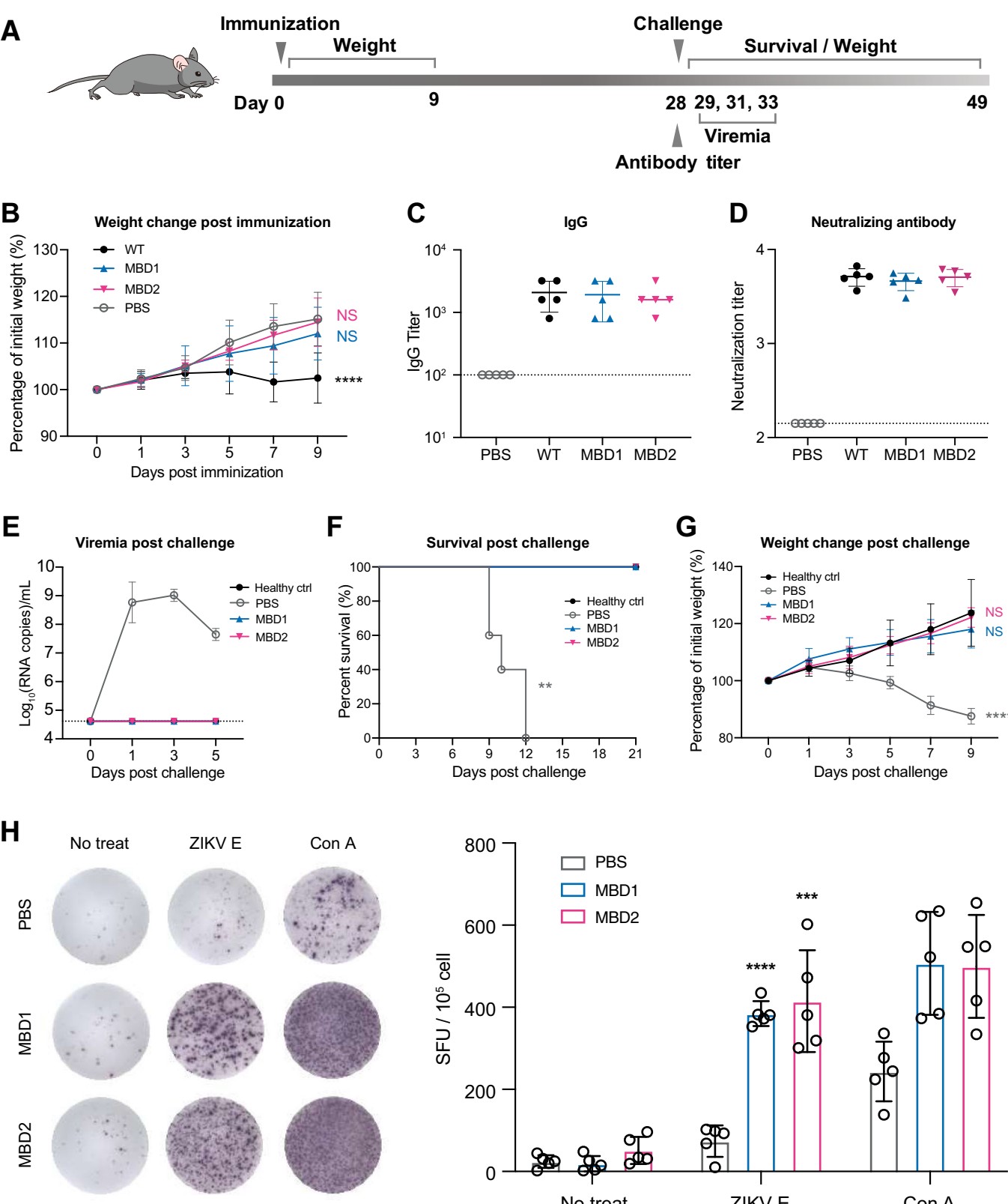

**Figure 5.  MBD ZIKV protects mice from WT ZIKV challenge.**

(A) Experimental scheme. Four-week-old A129 mice were immunized through the s.c. route with $1 \times 10^4$ PFU of MBD ZIKVs or PBS (negative control) ($n = 5$/group). On day 28 post-immunization, mice were challenged through i.p. route with $1 \times 10^5$ PFU of VEN/2016 ZIKV. (B) Weight changes on indicated day post-immunization, Data are the mean ± SD. $n = 5$. $n$ represents biological replicates. Two-way ANOVA, ****$P < 0.0001$ ($P = 1.95$e-007). (C) Anti-ZIKV IgG antibody titers in mouse serum on day 28 post-immunization. Data are the mean ± SD, $n = 5$. $n$ represents biological replicates. (D) Neutralizing antibody titers in mouse serum on day 28 post-immunization. The dashed line indicates the (LOD) of the assay. Data are the mean ± SD. $n = 5$. $n$ represents biological replicates. (E) Post-challenge viremia on indicated days was quantified by RT-qPCR. The dashed line indicates the (LOD) of the assay. Data are the mean ± SD. $n = 5$. $n$ represents biological replicates. (F, G) Survival (F) and weight changes (G) were monitored at indicated day post-challenge. (F) $n = 5$, $n$ represents biological replicates. log-rank test. **$P < 0.01$ ($P = 0.0025$). (G) Data are the mean ± SD. $n = 5$. Two-way ANOVA, ****$P < 0.0001$ ($P = 3.84$e-015). (H) T-cell responses in vaccinated mice were measured through ELISPOT assay. A129 mice were i.p. inoculated with $1 \times 10^4$ PFU of MBD ZIKVs or equal volume of PBS. ELISPOT assays specific for IFN-γ in splenocytes was performed on day 7 post-immunization. Representative images were shown in left panel and the spot forming units (SFU) were counted (right panel). Data are the mean ± SD. $n = 5$. $n$ represents biological replicates. Two-sided Student's $t$ test. ***$P < 0.001$, ****$P < 0.0001$ (MBD1 $P = 5.92$e-007, MBD2 $P = 0.0004$). Source data are available online for this figure.

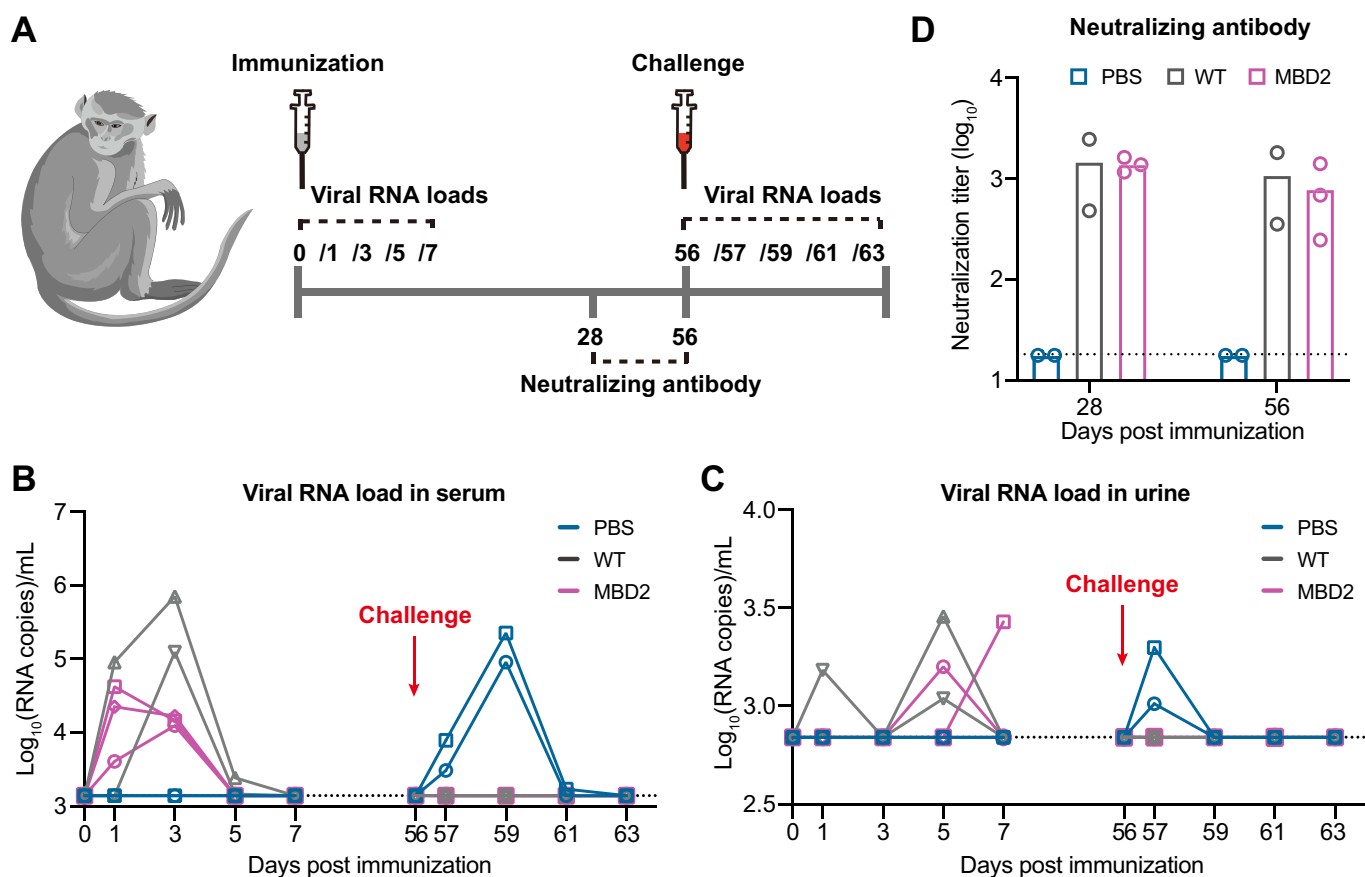

**Figure 6.  Immunogenicity and protection efficacy of MBD ZIKV in non-human primates.**

(A) Experimental scheme. Cynomolgus macaques were infected with $1 \times 10^5$ PFU of WT ZIKV ($n = 2$), MBD2 ($n = 3$) or PBS sham ($n = 2$) via the s.c. route. $n$ represents biological replicates. (B, C) Viral RNA loads in sera (B) and urine (C) were measured on indicated days post immunization by RT-qPCR. Each line represents data from an individual animal in each group. The dashed line indicates the LOD of the assay. (D) Antibody neutralization titers post immunization. Source data are available online for this figure.

JEV and YFV) by ELISA assay (Appendix Fig. S4). Then, cynomolgus monkeys were infected with WT ZIKV, MBD2 or PBS control by the s.c. route (Fig. 6A). As expected, both viruses produced robust viremia at 1–5 dpi, and the peak titers of MBD2 was lower than that from WT ZIKV (Fig. 6B). Meanwhile, discrete and low level viral RNA shedding was detected till 7 dpi in urine from WT ZIKV; while only one MBD2-immunized animal showed detectable viral RNA excretion at 5 dpi (Fig. 6C). Then, neutralizing antibodies against ZIKV were measured

at 28 and 56 dpi. As shown in Fig. 6D, MBD2 elicited high levels of neutralizing antibodies, which were comparable with that of WT ZIKV. As expected, no neutralizing antibodies was detected in the PBS-immunized animals.

At day 56 post immunization, all animals received MBD2 and PBS immunization were challenged with $10^5$ PFU of WT ZIKV. As expected, ZIKV challenge led to sustained viremia and urine shedding in the PBS-inoculated monkeys (Fig. 6B,C). In contrast,

no viral RNA was detected in the sera or urine of any MBD2-vaccined monkeys after challenge (Fig. 6B,C). These results clearly demonstrate that a single immunization with MBD2 induces protective immunity in non-human primates.

## Discussion

Here, we proposed and validated a novel strategy to achieve tissue-specific attenuation of ZIKV by disrupting the MBS in ZIKV genome. By leveraging the tissue-specific distribution of the host protein MSI1, we thoroughly profiled the replication kinetics and immunogenicity of recombinant MBD ZIKV in multiple organs in mouse and nonhuman primate models. The engineered MBD ZIKV is selectively attenuated in tissues vulnerable to ZIKV infection, while maintaining efficient replication in other tissues. This unique property ensures the safety of MBD ZIKV while enabling it to elicit robust and long-lasting humoral and cellular immunity, thereby providing effective protection against ZIKV challenge.

To date, several different kinds of strategies have been utilized to develop ZIKV LAV. One strategy is to develop chimeric LAVs that express ZIKV prM/E proteins within other licensed flavivirus vaccines. Compared to this approach, our MBD strategy targets viral UTR, allowing the preservation of the entire immunogenic antigens, thereby eliciting an immune profile of immune response identical to that of the WT virus. Indeed, the antibody titers induced by MBD ZIKV were comparable to those elicited by WT ZIKV (Figs. 5C,D and 6D). In addition to our strategy, other 3'UTR-targeted ZIKV LAV strategies have been proposed, including a 10-nucleotide deletion in the 3'UTR or mutations designed to disrupt the production of subgenomic flavivirus RNA (sfRNA) (Doets and Pijlman, 2024; Shan et al, 2017; Sparks et al, 2020). Compared to these approaches, our strategy achieves attenuation by disrupting the binding between viral RNA and the host protein MSI1. This allows the vaccine candidate to replicate efficiently in cell lines lacking MSI1 expression, as demonstrated by the fact that MBD ZIKV replicates efficiently in MSI1-negative Vero cells (Fig. EV3A,B), which are approved for vaccine production. This enables the simple and efficient production of high-titer vaccine stocks. Furthermore, our strategy targets the tertiary structure of viral RNA, requiring only minor nucleotide changes rather than large insertion or deletion. This makes the approach significantly less prone to reversion, as evidenced by the stability of MBD mutations even after ten rounds passages in Vero cells or four rounds of passages in the brains of neonatal mice (Appendix Fig. S3).

Our study reveals for the first time that specific host protein binding sites within viral genomes can be targeted for LAV development. Similar to MSI1, various RNA binding proteins (RBPs) play important roles in viral replication and pathogenesis. For instance, TRIM25, G3BP1, G3BP2 and CAPRIN1 binds to DENV2 3′UTR, facilitating the virus's evasion of the immune response (Bidet et al, 2014; Manokaran et al, 2015); IGF2BP1 binds the Hepatitis C virus (HCV) UTRs and SARS-CoV-2 RNA genome, promoting viral translation (Weinlich et al, 2009; Zhang et al, 2022). Targeting these RBP binding sites offers a promising strategy for designing LAVs. In addition to these pro-viral RBPs, some RBPs exert inhibitory effects on viral replication. For example, FMRP inhibits ZIKV replication by binding to its 3'UTR (Soto-Acosta et al, 2018); DDX39A inhibits alphaviruses infection through binding to a conversed RNA structure on the alphavirus genome RNA (Tapescu et al, 2023). These RBP binding sites could potentially be transplanted into other viruses to generate attenuated variants. Thus, this strategy of targeting host RBP binding sites hold potentials for application across a wide range of viruses.

Although the MBD ZIKV is highly attenuated in several animal models, the robust replication of MBD ZIKV in the MSI1-deficient tissues should not be neglected prior to any clinical trials. Further investigation about the long-term safety and efficacy of MBD ZIKV in different models should be warranted in the future. Another concern is cross-reactivity between different flavivirus members. Since antibody-dependent enhancement (ADE) has been observed among different flavivirus species (Castanha et al, 2017; Paul et al, 2016), the impact of ZIKV LAV immunization on subsequent DENV infection deserves further investigation. On the other hand, cellular immune responses induced by one flavivirus species have been shown to provide protection against other flaviviruses (Regla-Nava et al, 2018; Reynolds et al, 2018). It will be interesting to investigate whether our LAV approach can confer cross-protective immunity against other flaviviruses in future studies.

While our work has focused on MBD-mediated attenuation of LAV, the host RBP binding sites modification technology may also be applicable to other viral-based therapeutics. MSI1 is also highly expressed in a variety of tumors, such as glioblastoma, ovarian Adenocarcinoma and medulloblastoma (Chen et al, 2015; Uren et al, 2015; Vo et al, 2012). Oncolytic virus has been proven as a powerful weapon for tumor treatment (Chen et al, 2018), and insertion of MBSs into oncolytic viruses may enhance their tropism for tumors and improve their safety. Therefore, further development of the technology of RBP binding sites modification may have broad implications for the vaccine development and cancer therapy.

## Methods

**Reagents and tools table**

| Reagent/resource | Reference or source | Identifier or catalog number |
| --- | --- | --- |
| **Experimental models** | | |
| BHK-21 | ATCC | CCL10 |
| Vero | ATCC | CCL81 |
| A549 | ATCC | CRM-CCL-185 |
| SH-SY5Y | ATCC | CRL-2266 |
| Y79 | MeisenCTCC | CTCC-001-0343 |
| Bewo | MeisenCTCC | CTCC-400-0249 |
| Human cerebral organoid | This study | N/A |
| BALB/c | Charles River | N/A |
| CD-1 | Charles River | N/A |
| A129 | Laboratory Animal Center, Academy of Military Medical Sciences | N/A |
| Cynomolgus macaque | Laboratory Animal Center, Academy of Military Medical Sciences | N/A |
| **Viruses** | | |
| ZIKV strain FSS13025 | Shan et al (2016) | KU955593 |
| ZIKV strain VEN/2016 | Zhang et al (2016) | KU820898 |

| Reagent/resource | Reference or source | Identifier or catalog number |
| --- | --- | --- |
| MSI1 binding deficient (MBD) ZIKV mutants | Chen et al (2023) | N/A |
| **Antibodies** | | |
| MSI-1 | Abcam | ab52865 |
| ZIKV envelope protein | Biofront Technologies | BF-1176-46 |
| β-Actin | Abclonal | AC026 |
| α-Tubulin | Abclonal | AC012 |
| goat anti-rabbit IgG | Abcam | ab150080 |
| goat anti-mouse IgG | Abcam | ab150113 |
| goat anti-rabbit IgG Hrp | ZSGB-BIO | ZB-2301 |
| goat anti-mouse IgG Hrp | ZSGB-BIO | ZB-2305 |
| **Oligonucleotides and other sequence-based reagents** | | |
| ZIKV RNA primer | This study | F: GGTCAGCGTCCTCTCTAATAAACG R: GCACCCTAGTGTCCACTTTTTCC Probe: FAM-AGCCATGACCGACACCACACCGT-BQ1 |
| mMsi1 primer | This study | F: AGTTCGGGGAGGTGAAAGAG R: CTGTGCTCTTCGAGGAAAGG |
| mB2M primer | This study | F: CTCGGTGACCCTGGTCTTTC R: GGATTTCAATGTGAGGCGGG |
| mGAPDH primer | This study | F: AACTTTGGCATTGTGGAAGG R: ACACATTGGGGGTAGGAACA |
| hIFN-β primer | This study | F: TAGCACTGGCTGGAATGAGA R: TCCTTGGCCTTCAGGTAATG |
| hIFIT1 primer | This study | F: AGAAGCAGGCAATCACAGAAAA R: CTGAAACCGACCATAGTGGAAAT |
| hActin primer | This study | F: CCTGGCACCCAGCACAAT R: GCCGATCCACACGGAGTACT |
| **Chemicals, enzymes and other reagents** | | |
| DMEM | ThermoFisher | 11995-065 |
| RPMI medium 1640 | ThermoFisher | 22400-089 |
| F-12K | ThermoFisher | 21227-022 |
| DMEM/F-12 | ThermoFisher | 11320033 |
| Fetal bovine serum (FBS) | ThermoFisher | 10270106 |
| Dendronfluor kit | Histova | NEEP450 |
| DAPI | Sigma-Aldrich | D9542 |
| RIPA lysis buffer | Beyotime | P0013 |
| enhanced chemiluminescence (ECL) kit | ThermoFisher | 34580 |
| RNA loading buffer | Takara | 9169 |
| Hybond N+ nylon membrane | GE Healthcare | RPN303B |
| HRP-conjugated streptavidin | Beyotime | A0303 |
| 4% paraformaldehyde (PFA) | Biosharp | BL539A |
| PBS | VIVICUM | VCM3008 |
| Purelink RNA minikit | ThermoFisher | 12183018A |
| PrimeScriptTM RT-PCR kit | Takara | 064 A |
| TB Green® PrimeScript™ PLUS RT-PCR Kit | Takara | 096 A |
| coating buffer | Solarbio | C1055 |
| TMB | CWbio | CW0050S |

| Reagent/resource | Reference or source | Identifier or catalog number |
| --- | --- | --- |
| ZIKV E protein | Sino Biological | 40543-V08B4 |
| IFN-γ ELISPOT kit | MABTECH | 3321 |
| SuperScript III one-step RT–PCR kit | Invitrogen | 12574-026 |
| **Software** | | |
| GraphPad Prism (8.1) | https://www.graphpad.com | |
| ImageJ | https://imagej.nih.gov/ij/index.html | |
| R (4.0.3) | CRAN | |
| DESeq2 (1.30.1) | Bioconductor | |
| Pheatmap (1.0.13) | CRAN | |
| Ggplot2 (3.3.5) | Bioconductor | |
| **Other** | | |
| Olympus IX73 | Olympus | |
| SYNERGY HTX | BioTek | |
| BIOREADER® 7000 | BIO-SYS | |

## Viruses and cells

ZIKV strain FSS13025 (GenBank accession number KU955593) was originally isolated from a patient in Cambodia in 2010 (Shan et al, 2016). ZIKV strain VEN/2016 (GenBank accession number KU820898) was isolated from a patient returned from Venezuela in 2016 (Zhang et al, 2016). The MSI1 binding-deficient (MBD) ZIKV mutants were generated in our lab (Chen et al, 2023). The baby hamster kidney fibroblast (BHK-21) cells (ATCC CCL10), African green monkey kidney epithelial (Vero) cells (ATCC CCL81) were cultured in DMEM (Thermo Fisher Scientific) containing 7% fetal bovine serum (FBS, Biowest). A549 cells (ATCC CRM-CCL-185) were cultured in DMEM containing 10% FBS. SH-SY5Y cells (ATCC CRL-2266) were cultured using DMEM/F12 (Thermo Fisher Scientific) containing 10% FBS. The human retinoblastoma cells Y79 (CTCC-001-0343) were cultured in RPMI1640 (Thermo Fisher Scientific) containing 20% FBS. The human placenta choriocarcinoma cells Bewo (CTCC-400-0249) were cultured in F-12K (Thermo Fisher Scientific) containing 10% FBS. Experiments were randomized, and investigators were not blinded to the allocation process during the experiments and outcome assessments. The sample size was determined based on the known variability of each experiment.

## Histopathology and immunohistochemistry

For histopathology, testis tissues from mice were fixed in 4% neutral-buffered formaldehyde for 48 h, embedded in paraffin, sectioned, and stained with hematoxylin and eosin (H&E). Immunohistochemistry was performed using a Dendronfluor kit (NEEP450, Histova) following the manufacturer's instructions. Briefly, sections were deparaffinized with xylene, rehydrated through successive baths of ethanol/water (from 100% ethanol successively to 50% ethanol till pure water) and incubated in 3% $H_2O_2$ at room temperature. The sections were then put in 10 mM sodium citrate buffer for 1 h at 96 °C for antigen retrieval and blocked with BSA at saturation for 20 min. Indicated antibodies were applied to the section overnight at 4 °C. HRP-conjugated polyclonal antibody was used as a secondary antibody and TSA as a

## Western blot

The samples were lysed in lysis buffer (P0013, Beyotime) and separated by electrophoresis on 10% SDS-polyacrylamide gels, and the resolved proteins were transferred onto PVDF membranes. The membranes were blocked with 5% skimmed milk, and then probed with the corresponding primary antibodies (MSI1, 1:1500; Actin, 1:10,000; ZIKV E, 1:1000), and then washed four times for 5 min in PBS-T (P05B02, Gene-protein Link), followed by incubating with corresponding HRP-conjugated secondary antibodies (1:10,000), and then washed four times for 5 min in PBS-T. The signals were detected using an enhanced chemiluminescence (ECL) kit (34580, Thermo Fisher Scientific).

## Northern blot

Purified RNA was mixed with an equal volume of RNA loading buffer (9169, Takara), heated at 65 °C for 10 min, and immediately chilled on ice for 2 min. The samples were then separated by electrophoresis on a 6% urea-denaturing polyacrylamide gel at 150 V. Following electrophoresis, the RNA was transferred onto a Hybond N$^+$ nylon membrane (RPN303B, GE Healthcare) and crosslinked using UV light. The membrane was pre-hybridized in PerfectHyb Plus hybridization buffer (H7033, Sigma-Aldrich) for 30 min at 68 °C, then hybridized overnight (24 h) at the same temperature with a biotin-labeled RNA probe complementary to the full-length 3′ UTR region (nucleotides 10,380–10,808). After hybridization, the membrane was washed three times (10 min each) at 68 °C with 2× SSC containing 0.1% SDS, followed by three additional 10-min washes at room temperature with 0.5× SSC and 0.1% SDS. The membrane was then incubated for 1 h at room temperature in PBS containing 0.05% Tween-20, followed by another 1-h incubation with HRP-conjugated streptavidin (A0303, Beyotime). Afterward, it was washed four times (5 min each) with PBS containing 0.05% Tween-20. Signal detection was performed using an enhanced chemiluminescence (ECL) kit.

## Immunofluorescence analysis

Cells were fixed using BL539A for 15 min at room temperature (RT) and permeabilized using 0.2% Triton X-100 for 15 min at RT. The cells were then blocked in 3% bovine serum albumin for 2 h at 37 °C, and then incubated with the corresponding primary antibodies, and then washed with PBS three times for 5 min, followed by incubating with secondary antibodies at 37 °C for 1 h. After three washes with PBS, the cells were stained with DAPI for nuclei detection. The images were obtained from a microscope (Olympus IX73).

## Plaque assay

Virus samples were subjected to 10-fold serial dilutions in DMEM supplemented with 1% FBS. A volume of 500 µl from each dilution was applied to BHK-21 cells seeded in 12-well plates and incubated for 2 h at 37 °C in a 5% CO$_2$ atmosphere. After incubation, the supernatants were removed, and 1 mL of overlay medium—consisting of DMEM with 1% low-melting-point agarose (A600015, Sangon Biotech) and 2% FBS—was added to each well. Four days following infection, the cells were fixed with 4% formaldehyde at room temperature for 2 h, then stained using 1% crystal violet for 30 min. After thorough rinsing with water, plaque formation was visualized and counted to determine viral titers.

## RT-qPCR

Total RNA was extracted using Purelink RNA minikit (12183018 A, Thermo Fisher Scientific). To determine Zika viral RNA loads, One Step PrimeScript$^{TM}$ RT-PCR kit (RR064A, Takara) was used. Details of the primers used in this study are shown in the Reagents and Tools Table. To determine host mRNA levels, the One Step TB Green® PrimeScript™ PLUS RT-PCR Kit (RR096A, Takara) was used. Msi1 mRNA levels across mouse tissues were quantified and normalized to the geometric average of two reference genes, *B2m* and *GAPDH*. *IFN-β* and *IFIT1* mRNA levels were quantified and normalized to the reference gene Actin. Details of the primers used in this study are shown in the Reagents and Tools Table. The ΔΔCT values were determined using control samples as the reference value. Relative levels of mRNAs were calculated using the formula $2^{(-\Delta\Delta CT)}$.

## Brain organoid formation and infection

Brain organoid formation was performed based on a previously described protocol (Lancaster et al, 2013). Human pluripotent stem cells (hPSCs) were seeded in low-attachment 96-well plates at 9000 cells per well and cultured in embryoid body (EB) formation medium at 37 °C with 5% CO$_2$. On day 5, the medium was replaced with fresh medium containing 20 ng/mL bFGF. On day 6, EBs were transferred to 24-well plates and cultured in neural induction medium for 5 days, with media changes every other day. On day 10, the aggregates were embedded in Matrigel for 30 min at 37 °C, then transferred to early differentiation medium in 6-well plates and cultured for 3 days. On day 13, cultures were switched to organoid maturation medium. Viral infection was carried out on day 15. Viral RNA levels were quantified, and images were acquired as described in the previous methods. Organoid areas were measured using ImageJ software.

## Enzyme-linked immunosorbent assay

The IgG antibody titers in the animal sera were determined by Enzyme-linked immunosorbent assay (ELISA). For the ZIKV IgG test of the mouse sera in Fig. 5B, ELISA plates were coated with recombinant ZIKV E protein (1 mg/ml) (40543-V08B4, Sino Biological) in a coating buffer (C1055, Solarbio). For the flaviviruses IgG test of monkey sera prior of vaccination in Appendix Fig. S4, ELISA plates were coated with 10$^4$ PFU of heat-inactivated virus per well. The plates were coated overnight at 4 °C. Subsequently, the coated plates were incubated with serially diluted serum for 1 h at 37 °C. Plates were washed for three times with PBS-T. After incubation with HRP-conjugated secondary antibody (1:5000) for 1 h at 37 °C. After three washes in PBS-T, the plates were incubated with 3,3′,5,5′-tetramehylbenzidine substrate (TMB) (CW0050S, CWbio). Then 2 M H$_2$SO$_4$ was added

to stop the reaction. The absorbance (450 nm) was read using a microplate reader (SYNERGY HTX, BioteK). ELISA endpoint titers were defined as the highest reciprocal serum dilution that yielded an absorbance exceeding twofold over background values.

## Neutralization assay

Fifty percentage plaque reduction neutralization test ($PRNT_{50}$) was used to determine the neutralizing antibody titers. Briefly, serum samples were heated for 30 min at 56 °C for inactivation. Then the inactivated sera were fourfold diluted serially. These dilutions of serum were mixed with an equal volume of ZIKV solution, resulting in a mixture containing approximately 250 PFU of virus. After incubation at 37 °C for 1 h, the viral titers of the mixtures were determined by the plaque forming assay using BHK-21 cells The $PRNT_{50}$ titers were calculated by the method of Spearman-Karber (Hamilton et al, 1977).

## Enzyme-linked immunospot (ELISPOT)

IFN-γ ELISPOT assay was performed using the commercial mouse IFN-γ ELISPOT kit (3321, MABTECH) following the manufacturer's instructions. Briefly, freshly isolated splenocytes from MBD ZIKV immunized mice were plated at $5 \times 10^5$ cells per well in presence of recombinant ZIKV E protein (0.4 μg/ml) (40543-V08B4, Sino Biological) or medium alone (negative control) or 10 μg/ml of Concanavalin A (Con A) (Con A was used as a positive control, as its binds to T cell surface glycoproteins and activates T cell and the cytokine production). After incubation at 37 °C, 5% $CO_2$ for 48 h, the plates were washed and incubated with biotinylated anti-IFN-γ antibody for 1 h at 37 °C, followed by incubation with streptavidin-ALP for 1 h at room temperature. The IFN-γ spots were developed by adding substrate solution. After drying, the spots were scanned and counted using ELISPOT image analysis (BIOREADER® 7000, BIO-SYS).

## Ethical statement

Research procedures involving animals were conducted according to ethical guidelines and approved by the Institutional Animal Care and Use Committee (IACUC) of Military Medical Sciences (Approval number. IACUC-IME-2021-027).

## Mouse studies

BALB/c and CD-1 mice used in this study were purchased from Beijing Vital River Laboratory Animal Technology. A129 mice were purchased from Laboratory Animal Center, Academy of Military Medical Sciences. All mice were received standard chow and purified water ad libitum. Environmental controls maintained a 12:12-h light/dark cycle, temperature at $22 \pm 1$ °C, and relative humidity at $55 \pm 5\%$. For neurovirulence test, one-day-old suckling BALB/c mice were injected with $1 \times 10^3$ PFU of WT ZIKV, MBD1 or MBD2 through intracerebral (i.c.) route and observed for survival within 21 days. Survival analysis was performed using the GraphPad Prism software. To test the replication of MBD ZIKV in mice, 4-week-old male A129 mice were infected s.c. with $1 \times 10^4$ PFU of WT ZIKV, MBD1 or MBD2. Multiple organs were

collected for viral RNA loads and viral titers determination at 7 dpi.

For immunogenicity test, 4-week-old A129 mice were inoculated s.c. with $1 \times 10^4$ PFU of MBD1 and MBD2. The first-passage (P1) virus from C6/36 cells was used for vaccination. The mice were bled by tail vein puncture 1 day before (Day 0) and at 4 weeks post immunization. The IgG and neutralizing antibodies in the sera were determined by ELISA and $PRNT_{50,}$ respectively. Immunized mice were i.p. challenged with $1 \times 10^5$ PFU of ZIKV strain VEN/2016 at 4 weeks post immunization. Viral RNA loads in sera were determined at day 0, 1, 3 and 5 after challenge.

To test the vertical transmission of ZIKV, CD-1 pregnant dams were injected i.p. with 2 mg of mouse anti-IFNAR1 antibody (clone MAR1-5A3; Leinco Technologies) on E5.5. On E6.5, mice were injected s.c. with $1 \times 10^5$ PFU of ZIKV. Viral RNA loads in serum at days 1–3 post-challenge were determined. Mice were killed on E13.5 and placentas, fetuses, and maternal tissues were collected for viral RNA loads determination.

## RNA-seq and data analysis

One-day-old suckling CD-1 mice ($n = 3$/group) were i.c. inoculated with 1000 PFU of indicated viruses. On days 3 post inoculation, total RNA from brains were extracted using TRIzol (Invitrogen, Carlsbad, CA, USA) and DNase I (NEB, USA) treated, respectively. Sequencing libraries were generated using FCL PE150 V3.0 RNA Library Prep Kit for DNBSEQ T7RS following the manufacturer's recommendations. After sequencing, Perl script was used to filter the original data (Raw Data) to clean reads by removing contaminated reads for adapters and low-quality reads. Clean reads were aligned to the Mus musculus genome using Hisat2. The number of reads mapped to each gene in each sample was counted by HTSeq and TPM was then calculated to estimate the expression level of genes in each sample. Differentially expressed genes (DEGs) were defined as those with $P$adj <0.05 and |Log2FC| ≥1. DEGs were employed as a query to search for enriched biological processes (Gene Ontology BP) using Metascape. Heatmaps of gene expression levels were constructed using pheatmap package in R. Dot plots, volcano plots and bar plots were constructed using ggplot2 package in R.

## Stability of MBD mutants

To assess the in vitro stability of MBD mutants, the viral stock was used to infect Vero cells (MOI = 0.1) in a well of a six-well plate containing a confluent monolayer of cells. At 3 dpi, 100 μL of culture fluid was transferred to a new well of six-well plate containing naive Vero cells. After ten rounds of such passaging, the viral RNAs from culture fluid was extracted as described above. cDNA synthesis and PCR amplification of the viral RNAs were performed using the SuperScript III one-step RT–PCR kits (12574-026, Invitrogen). The PCR products were subsequently subjected to sequencing.

For in vivo stability assessment of MBD mutants, one-day-old suckling CD-1 mice were injected with $1 \times 10^3$ PFU of MBD1 or MBD2 through intracerebral (i.c.) route. At 3 dpi, the infected brains were homogenized with 500 μL DMEM, followed by centrifugation. The supernatant was collected, and 10 μL of

**The paper explained**

**Problem**

Although viral RNA secondary/tertiary structures are recognized for multifunctional roles, including specific host protein binding, their utility in LAV design has been minimally investigated.

**Results**

Disruption of the primary sequence or tertiary configuration of a Zika virus (ZIKV) RNA domain essential for Musashi-1 (MSI1) binding was shown to produce tissue-selective attenuation across animal models. Engineered MSI1-binding-deficient ZIKV mutants (MBD) retained full infectivity in MSI1-absent tissues but displayed severe restriction in vulnerable sites (brain, testis, eye, placenta) and markedly diminished vertical transmission in mice. A single MBD immunization elicited potent immune responses and provided complete protection against ZIKV challenge in both murine and non-human primate systems.

**Impact**

This work validates targeted manipulation of host protein-interacting viral RNA structures as a robust foundation for pioneering next-generation LAV platforms against emerging viral threats.

supernatant was used to infect a new set of one-day-old suckling CD-1 mice via the i.c. route. After four rounds of such passaging, the viral RNAs was extracted from the infected brains and sequenced as described above.

## Monkey experiments

Cynomolgus macaques were purchased from Laboratory Animal Center, Academy of Military Medical Sciences. All non-human primates were housed individually in separate cage under controlled conditions. Rooms maintained a 12:12-h light/dark cycle, temperature at $24 \pm 2$ °C, and humidity at $60 \pm 10\%$. Animals received standard diet twice daily, supplemented with fresh fruits/vegetables, with ad libitum access to purified water. For immunization and challenge of cynomolgus macaques, a total of seven cynomolgus monkeys (4–5 years old, weighing 2.5–5.0 kg). Monkeys were inoculated s.c. with 0.5 ml of viral solution containing $1 \times 10^5$ PFU of WT ZIKV ($n = 2$) or MBD2 ZIKV ($n = 3$), or sham PBS ($n = 2$). The first-passage (P1) virus from C6/36 cells was used for vaccination. Clinical signs were monitored for 10 days after inoculation. Blood and urine were collected at 0, 1, 3, 5 and 7 dpi to detect viral RNA loads. The neutralizing-antibody titers of blood samples at 28 and 56 dpi were determined by $PRNT_{50}$ as described above. On day 56 dpi, the animals immunized with MBD2 ZIKV and the PBS controls were s.c. challenged with $1 \times 10^5$ PFU of the ZIKV strain FSS13025. For the following 7 days, blood and urine were collected for the determination of viral RNA loads.

## Statistical analysis

All data were analyzed using GraphPad Prism (version 8.1). Unpaired two-sided Student's $t$ test, two-way ANOVA with Bonferroni correction or log-rank test were used as indicated in the Figure legends. Number of biological replicates or animals ($n$) used are stated in the Figure legends. Data are presented as mean ± standard deviation (SD).

## Data availability

The datasets produced in this study are available in the following databases: RNA-Seq data: Gene Expression; Omnibus GSE301761.

The source data of this paper are collected in the following database record: biostudies:S-SCDT-10_1038-S44321-025-00304-5.

## Peer review information

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

## Acknowledgements

We thank Prof. X-Y Fang and Dr. Y Wang for helpful discussion. C-FQ was supported by the National Key Research and Development Program of China (No. 2021YFC2302400 and 2023YFC2305900), and the Innovation Fund for Medical Sciences (No. 2019RU040) from the Chinese Academy of Medical Sciences (CAMS). XC was supported by the National Natural Science Foundation of China (No. 82402132).

## Author contributions

**Xiang Chen**: Conceptualization; Data curation; Formal analysis; Funding acquisition; Validation; Investigation; Visualization; Methodology; Writing—original draft; Writing—review and editing. **Meng-Li Cheng**: Resources; Investigation; Methodology. **Xing-Yao Huang**: Resources; Investigation; Methodology. **Meng-Xu Sun**: Resources; Investigation; Methodology. **Rui-Ting Li**: Resources; Data curation; Formal analysis; Investigation; Methodology. **Mei Wu**: Resources; Investigation. **Yu-Yan Li**: Formal analysis; Investigation. **Qian Xu**: Investigation. **Meng-Yue Guan**: Investigation. **Hui Zhao**: Supervision. **Cheng-Feng Qin**: Conceptualization; Resources; Supervision; Funding acquisition; Project administration; Writing—review and editing.

Source data underlying figure panels in this paper may have individual authorship assigned. Where available, figure panel/source data authorship is listed in the following database record: biostudies:S-SCDT-10_1038-S44321-025-00304-5.

## Disclosure and competing interests statement

C-FQ and XC have filed patents related to the finding reported in this manuscript. The remaining authors declare no competing interests.

# Expanded View Figures

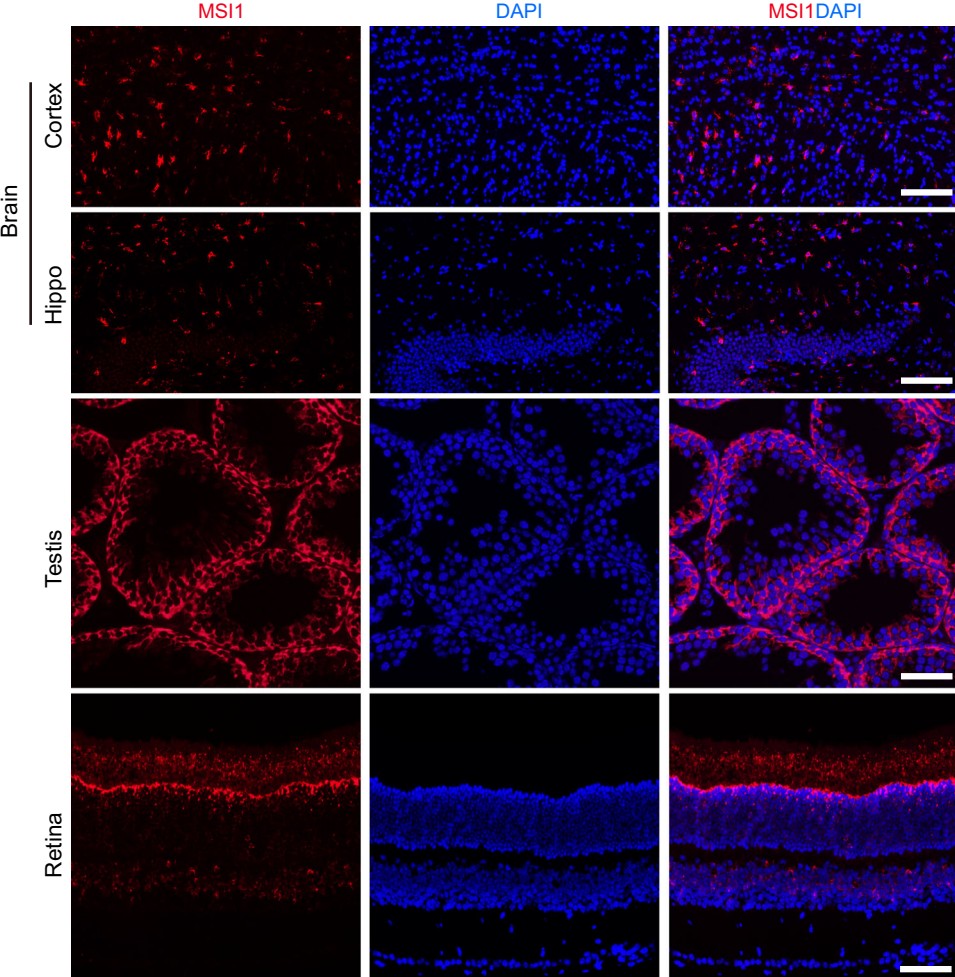

**Figure EV1. Characterization of MSI1 expression in tissues vulnerable to ZIKV.**

Brain, testis and retina sections of 4-week-old A129 mice were stained with an anti-MSI1 antibody. Scale bar, 100 μm.

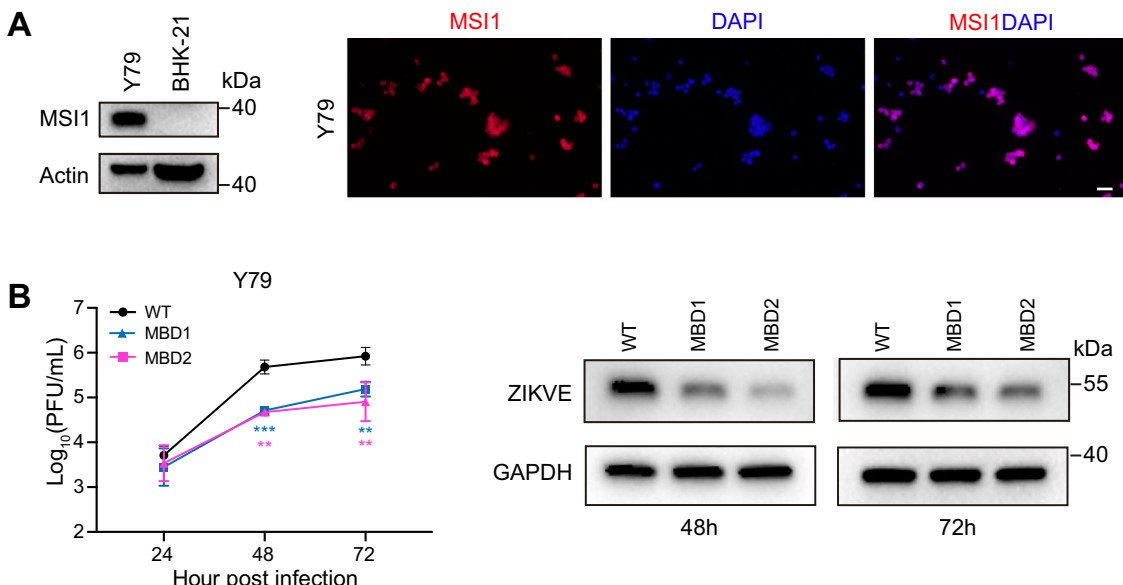

**Figure EV2.  MBD ZIKV exhibits attenuated replication kinetics in a human retina cell line expressing MSI1.**

(A) MSI1 expression in human retinoblastoma Y79 cells was detected by Western blotting (left panel) and immunostaining (right panel). Scale bar, 50 μm. (B) Y79 cells were infected with WT ZIKV, MBD1 or MBD2 (MOI = 1), and the culture supernatants were harvested at the indicated time points for detection of viral loads by plaque forming assay (left panel). Data are mean ± SD. $n = 3$. $n$ represents biological replicates. Two-way ANOVA, **$P < 0.01$, ***$P < 0.001$ (48: MBD1 $P = 0.0007$, MBD2 $P = 0.0023$; 72: MBD1 $P = 0.0058$, MBD2 $P = 0.0021$). The expression of ZIKV-E protein at 48 h and 72 h after infection was detected by Western blotting (right panel).

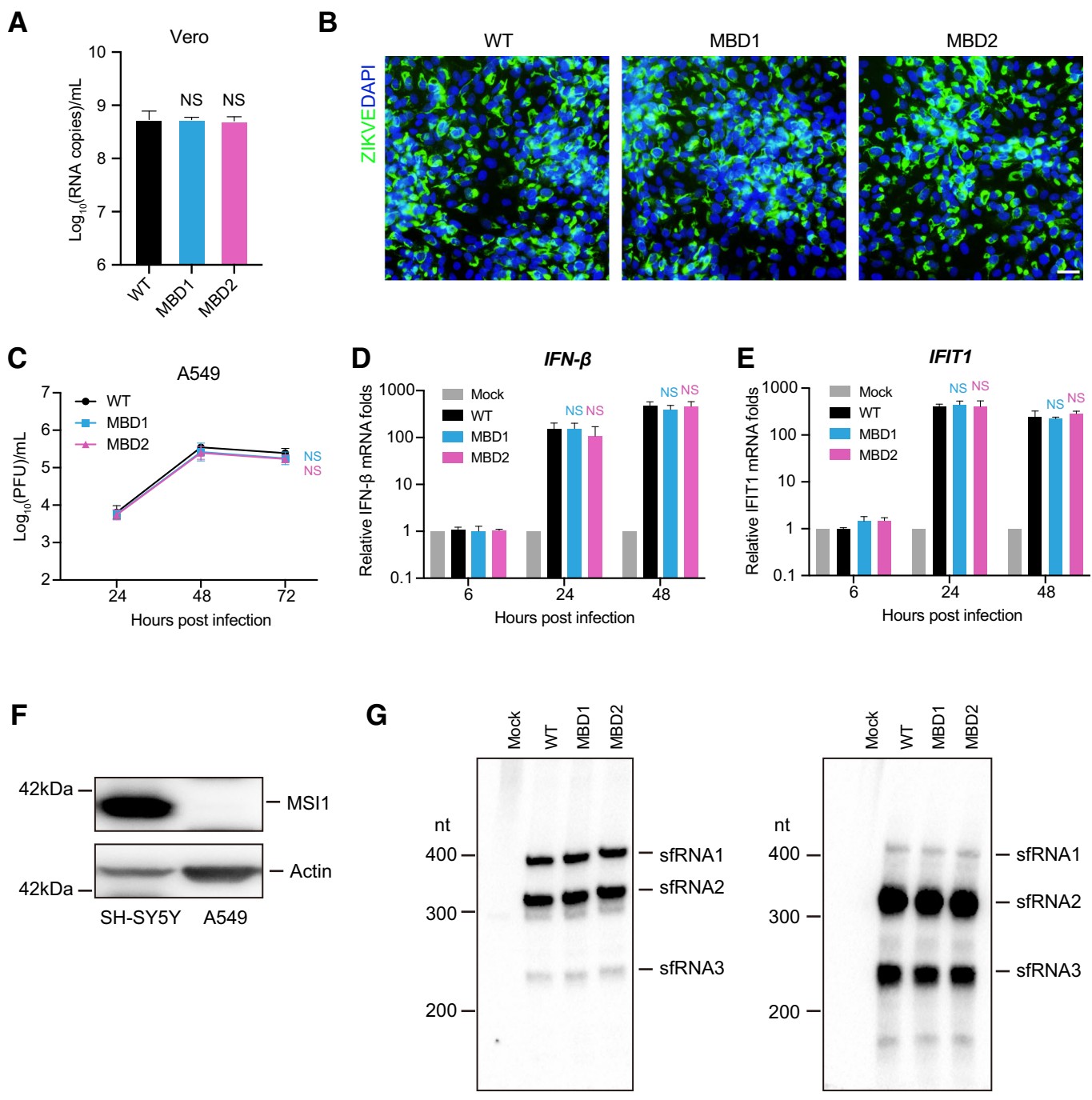

**Figure EV3. The attenuation of MBD ZIKV is independent of innate immune or sfRNA.**

(A) Vero cells were infected with WT ZIKV, MBD1 or MBD2 (MOI = 0.1), and the viral RNA loads in culture supernatants at 72 h after infection were detected by RT-qPCR. Data are mean ± SD. n = 3. n represents biological replicates. Two-sided Student's t test. NS, not significant. (B) The expression of ZIKV-E protein at 72 h after infection was detected by immunostaining. Scale bar, 50 μm. (C) A549 cells were infected with WT ZIKV, MBD1 or MBD2 (MOI = 0.1), and the culture supernatants were harvested at the indicated time points for detection of viral loads by plaque forming assay (left panel). Data are mean ± SD. n = 3. n represents biological replicates. Two-way ANOVA, NS not significant. (D, E) Relative levels of IFN-β (D) and IFIT1 (E) mRNA of infected A549 cells at indicated times after infection determined by RT-qPCR. The value of mock infected at 6 h post infection was set as 1. Data are the mean ± SD. n = 3. n represents biological replicates. Two-sided Student's t test. NS not significant. (F) The expression of MSI1 protein of SH-SY5Y and A549 cells was detected by Western blotting. (G) The sfRNA expression in SH-SY5Y and A549 cells after infection was detected by Northern blotting.

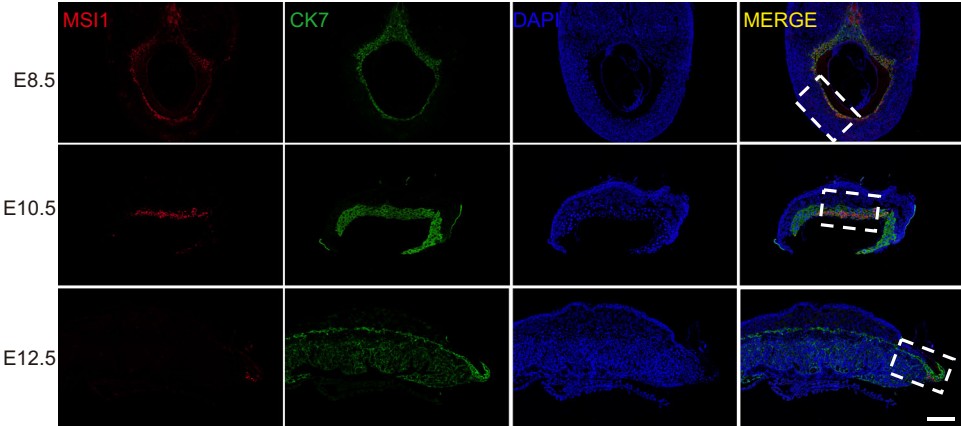

**Figure EV4. Characterization of MSI1 expression in the placenta during mouse pregnancy.**

Overview of immunostaining of mouse placenta sections at embryonic stage 8.5 (E8.5), E10.5, and E12.5. Tissue sections were stained with anti-MSI1(red) antibody and anti-CK7 antibody (green), a trophoblast cell marker. Nuclei were stained with DAPI (blue). The boxed areas are shown at higher magnification in Fig. 4A. Scale bar, 500 μm.

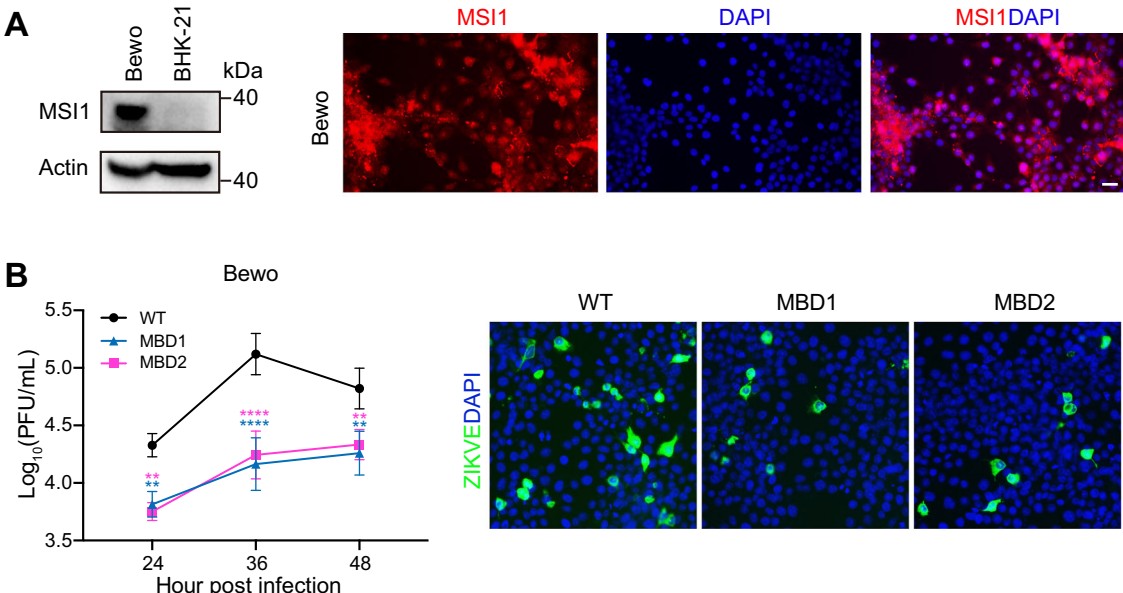

**Figure EV5.  MBD ZIKV exhibits attenuated replication kinetics in a human placental cell line expressing MSI1.**

(A) MSI1 expression in human placenta choriocarcinoma Bewo cells was detected by Western blotting (left panel) and immunostaining (right panel). Scale bar, 50 μm. (B) Bewo cells were infected with WT ZIKV, MBD1 or MBD2 (MOI = 1), and the culture supernatants were harvested at the indicated time points for detection of viral loads by plaque forming assay (left panel). Data are mean ± SD. $n = 3$. $n$ represents biological replicates. Two-way ANOVA, **$P < 0.01$, ****$P < 0.0001$ (24: MBD1 $P = 0.0017$, MBD2 $P = 0.0093$; 36: MBD1 $P = 0.000041$, MBD2 $P = 0.000052$; 48: MBD1 $P = 0.0062$, MBD2 $P = 0.0049$). The expression of ZIKV-E protein at 36 h after infection was detected by immunostaining (right panel). Scale bar, 50 μm.

