## [Peer Review File · EMBO Molecular Medicine]

Manipulating Zika virus RNA tertiary structure for developing tissue-specific attenuated vaccines

Xiang Chen, Meng-li Cheng, Xing-Yao Huang, Meng-Xu Sun, Mei Wu, Yu-Yan Li, Qian Xu, Rui-Ting Li, Meng-Yue Guan, Hui Zhao, and Cheng-Feng Qin

Corresponding author: Cheng-Feng Qin (qinfc@bmi.ac.cn)

Review Timeline:

Submission Date:	3rd Mar 25
Editorial Decision:	15th Apr 25
Revision Received:	18th Jul 25
Editorial Decision:	15th Aug 25
Revision Received:	18th Aug 25
Accepted:	20th Aug 25

Editor: Zeljko Durdevic

Transaction Report:

15th Apr 2025

Dear Prof. Qin,

Thank you for the submission of your manuscript to EMBO Molecular Medicine, and please accept my apologies for the unusual delay in getting back to you. We have now received feedback from two of the three reviewers who agreed to evaluate your manuscript. As the referee #3 will unfortunately not be able to return his/her report in a timely manner, and given that both reviewers provide very similar recommendations, we prefer to make a decision now in order to avoid further delay in the process. Should referee #3 provide a report, we will send it to you, with the understanding that we will not ask for an additional revision.

As you will see from their reports pasted below, both referees recognize potential interest of the manuscript but also raise serious concerns that should be addressed in a major revision. If you would like to discuss further the points raised by the referees, I am available to do so via email or video. Let me know if you are interested in this option.

We would welcome the submission of a revised version within three months for further consideration. Please let us know if you require longer to complete the revision.

I look forward to receiving your revised manuscript.

Yours sincerely,

Zeljko Durdevic

Zeljko Durdevic
Senior Editor
EMBO Molecular Medicine

We require:

- 1) A .docx formatted version of the manuscript text (including legends for main figures, EV figures and tables). Please make sure that the changes are highlighted to be clearly visible.
- 2) Individual production quality figure files as .eps, .tif, .jpg (one file per figure). For guidance, download the 'Figure Guide PDF': (<https://www.embopress.org/page/journal/17574684/authorguide#figureformat>).
- 3) A .docx formatted letter INCLUDING the reviewers' reports and your detailed point-by-point responses to their comments. As part of the EMBO Press transparent editorial process, the point-by-point response is part of the Review Process File (RPF), which will be published alongside your paper.
- 4) A complete author checklist, which you can download from our author guidelines (<https://www.embopress.org/page/journal/17574684/authorguide#submissionofrevisions>). Please insert information in the checklist that is also reflected in the manuscript. The completed author checklist will also be part of the RPF.

6) It is mandatory to include a 'Data Availability' section after the Materials and Methods. Before submitting your revision, primary datasets produced in this study need to be deposited in an appropriate public database, and the accession numbers and database listed under 'Data Availability'. Please remember to provide a reviewer password if the datasets are not yet public (see <https://www.embopress.org/page/journal/17574684/authorguide#dataavailability>).

.

- the medical issue you are addressing,

- the results obtained and

- their clinical impact.

12) Author contributions: You will be asked to provide CRediT (Contributor Role Taxonomy) terms in the submission system. These replace a narrative author contribution section in the manuscript.

13) A Conflict of Interest statement should be provided in the main text.

14) Every published paper now includes a 'Synopsis' to further enhance discoverability. Synopses are displayed on the journal webpage and are freely accessible to all readers. They include a short stand first (maximum of 300 characters, including space) as well as 2-5 one-sentences bullet points that summarizes the paper. Please write the bullet points to summarize the key NEW findings. They should be designed to be complementary to the abstract - i.e. not repeat the same text. We encourage inclusion of key acronyms and quantitative information (maximum of 30 words / bullet point). Please use the passive voice. Please attach these in a separate file or send them by email, we will incorporate them accordingly.

15) Include a Reagents and Tools Table as part of the Methods section, which can be downloaded from our author guidelines (<https://www.embopress.org/page/journal/17574684/authorguide#structuredmethods>)

***** Reviewer's comments *****

Referee #1 (Remarks for Author):

In this manuscript, the authors proposed a novel live-attenuated vaccine strategy by targeting the tertiary structure of ZIKV viral RNA that interacts with a specific host protein Musashi1 (MSI1). The authors designed and engineered two mutants, MBD1 and MBD2, for vaccine immunogenicity and safety assays. MBD ZIKV mutants exhibit restricted replication in MSI1-expressing tissues while maintaining robust replication in MSI1-deficient cells (e.g., Vero cells). A single immunization of MBD ZIKV induced strong humoral and cellular immunity, protecting mice and non-human primates (NHPs) against wild type ZIKV challenge. The introduced mutations remained intact after serial passages in vitro and in vivo. Overall, this present study provides a proof-of-concept validation for the use of viral RNA-protein interactions as a rational attenuation strategy, and similar strategies may be applicable to other viruses.

The experiments are well designed, and the manuscript is clearly written. Nevertheless, there are few concerns that the authors need to address for the manuscript to warrant publication.

1) The MSI1 expression patterns are the critical determinants for vaccine design. The major findings of the present manuscript were based on mouse model (partially validated in monkey), is there any difference in MSI1 expression in human or during development?

2) The authors proposed the tertiary structure of MBS2 mediates MSI1 binding, and the MBD2 mutant was generated by disrupting this tertiary structure. To strengthen this mechanistic claim, functional validation is required to demonstrate that MSI1 binding to MBS2 is structurally dependent (i.e., conformation-sensitive) rather than solely sequence-dependent. This distinction is critical for establishing the specificity of the attenuation mechanism.

3) Flavivirus xrRNAs (including MBS-containing elements) are known to perform multiple regulatory functions in viral replication. The authors are suggested to investigate whether MBD mutations affect other functions (e.g., sRNA production) to rule out off-target effects.

4) While the manuscript highlights the broad applicability of this strategy, it would be beneficial for the authors to provide a prioritized list of specific viral candidates

5) Figure.5: Did any adverse events (e.g., morbidity/mortality) occur during the immunization procedure?

6) Figure.4G: were there any difference in the weight of fetal heads between the two groups?

Referee #2 (Comments on Novelty/Model System for Author):

This manuscript was enjoyable to read and is relevant considering that there are no available vaccines against Zika virus. The results are convincing. The attenuated strain (strikingly and interestingly) show no disease in mice although they replicate only relatively slightly less than the parental strain.

However, as detailed in my comments below, the proposed Musashi1-dependent mechanism behind this attenuation is not convincing. In particular, there is a possibility that the modification of the 3'UTR disrupts the capacity of the virus to interfere with

the immune system (in addition to modulating replication efficiency), which would explain the phenotypes. This is properly not addressed here as well as in their previously published study). Several experiments are proposed in my remarks to the authors below.

The authors also need to better justify why these ZIKV mutants would be suitable attenuated vaccines as the parental strain is already not pathogenic and was excluded from the mice immunization/challenge study.

This is a nice study and addressing these concerns, would make this study suitable for publication EMBO Mol Med.

Referee #2 (Remarks for Author):

In this manuscript, Chen and colleagues developed mutants of the Zika virus (ZIKV) by disrupting RNA motifs in the viral genome, which bind to the host protein Musashi-1 (MSI1). The same research group had previously shown that these RNA structures play a role in viral replication in MSI1-expressing cells. The authors generated the MBD1 mutant with a single mutation changing the motif, and MBD2 again with a single mutation in 3' UTR region of ZIKV RNA changing in that case the tertiary structure of a tetraloop. This study shows that the use of these attenuated mutants with structural changes to of Musashi-1 binding site (MBS) potentially offers an immune protection in both mice and monkey challenge infection models. Furthermore, they demonstrate that in tissues where ZIKV infection causes more severe pathology, such as in neural, placental, and testicular tissues, these mutants replicate less than the parental strain, strikingly correlating with an abrogation of the pathogenesis. This allowed the authors to claim that the perturbation of MBS led to the absence of MSI1 binding in these MSI1 expressing tissues, which is necessary for proper viral replication. As a result, these mutants did not replicate efficiently, and the authors proposed that they could be safely administered as a vaccine.

The study highlights the significant potential of modifying the 3' UTR region of the viral genome to develop live attenuated virus vaccines. The data demonstrating tissue-specific responses to these modified sequences, driven by the ability of specific host proteins to bind these regions and modulate viral replication, further strengthens the appeal of the study concept. The *in vivo* phenotypes in terms of virus-induced symptoms are very convincing. However, some of the results especially regarding the mechanisms behind attenuation must be strengthened to make the conclusions more convincing. One concern is that the study's claim of tissue-specific viral attenuation is primarily based on RT-qPCR readouts, which may not always accurately reflect the actual efficacy of viral replication. Titration of infectious viral particles (using for instance TCID₅₀, focus-forming unit or plaque assays), at least for some key experiments (e.g., brain samples), would be highly beneficial in further supporting the attenuation phenotype of these mutants and help better assess the difference between WT and mutant ZIKV. The same applies for *in vitro* experiments. Since RT-qPCR also detect dead cells and not necessarily RNA active in replication (e.g. degradation products), viral titers (reflecting a productive viral replication cycle) might actually show in the end that attenuation is more pronounced than what is concluded here (only 10-fold with the RNA readout).

The authors also assessed the impact of two ZIKV vaccine strains on one of the most severe ZIKV-associated complications, namely microcephaly, in a neonatal mouse model. They found that infection in the brain with low MSI1 expression, using the MD1 and MD2 strains, did not induce microcephaly, showing a 1-log reduction in viral RNA and the absence of the microcephaly phenotype. This complete restoration of survival and neonatal brain size with the mutants is very convincing. However, with only one Log₁₀ reduction in viral RNA levels, it is hard to comprehend how such relatively mild attenuation can lead to such drastic phenotypes. This raises doubts that this is only due to the replication capacity of these mutant viruses and further suggest that other mechanisms are at play. Given that neuroinflammation can drive developmental changes, a more detailed analysis would strengthen the claim that the absence of microcephaly is due to reduced viral load rather than immune modulation. Additional data on the immune response, including interferons and key proinflammatory cytokines/chemokines with both wt and mutant viruses, would clarify whether the phenotypes associated with the mutants result from immune modulation or viral load reduction.

The discrepancy between the differences in viral loads and phenotypes *in vivo* raises a fundamental question about UTR modifications and their potential impact on sfRNA status, which can influence the immune response as well as virus-induced cell death. First, examining sfRNA status in human MSI1-negative cells could further support their hypothesis like in their previous work with northern blots in hamster BHK-21 cells showing no impact on sfRNA expression profile. However, this must be done in immune-competent human cells (ideally in both MSI1-expressing and MSI1 deficient cells). This would help demonstrate that the viral attenuation of these mutants is not dependent on sfRNA expression. In the same line of idea, the authors show replication in MSI1-deficient Vero cells (Fig. S3) and use these data to claim that the attenuation is MSI1-dependent. This is not ideal because these cells are not human, do not have a functional innate immune system and are extremely efficient at producing viruses. Here, only the 72 hpi time point at which production is most probably saturated. To further support their claim, the authors must perform viral replication kinetics in human cells lacking MSI1 but still competent in MAVS-dependent innate immunity which would provide stronger evidence that attenuation is MSI1-dependent. Additionally, assessing immune response (gene or protein) in this context would strengthen the overall findings.

This study suggests that the ZIKV mutant's vertical transmission capability is restricted. The authors first analyze the expression levels of MSI1 in the placenta and hypothesize that the presence of MSI1 in placenta may play a role in vertical transmission. In conclusion, they observe lower viral loads in the fetal heads and placenta of pregnant mice infected with MSD2 strain compared to the WT virus and base their claim of reduced transmission on this observation. The author only analyzed the maternal spleen and refer to results with A129 mice, which lack immunity compared to CD1 animals. To firmly conclude that vertical transmission

is affected, further investigation across multiple organs is needed, particularly with those where no significant differences between wild-type and mutant strains are observed. This would help clarify that the reduced viral load in the placenta and fetal heads of CD1 mice infected with MBD2 is not simply due to a lower amount of virus in the maternal periphery. The authors might consider measuring viral loads in the serum/blood to provide valuable insight as to whether there is generally less circulating virus in peripheral organs at this stage. Lastly, it would be helpful if the authors could clarify why MSD2 was prioritized for the vertical transmission study as the MSD1 mutant was omitted despite resulting in similar results in other experiments.

The authors performed the entire study using the pre-epidemic (and presumably non-pathogenic) FSS13025 strain (Cambodia 2010). Did the authors consider testing more recent, microcephaly-associated strains in their models, such as H/PF/2013 or Brazilian strains? It would be highly relevant to assess whether prior immunization with FSS13025 MSD2 potentially offers protection against these more aggressive strains. Importantly, in Figure 5, the parental wt strain was not included in the analysis. In that context, it is really hard to judge what is the advantage of considering the mutants as vaccine strains compared to the wt since the latter replicates only a bit better (approximately 1 Log₁₀ as assessed by RT-qPCR) and is presumably not pathogenic in Human.

Other comments:

- The authors regularly use the word "titer" when measuring the viral RNA load (e.g. line 133). This is not correct since it does not necessarily correlate with the abundance of infectious particles and a productive and complete ZIKV life cycle. RT-qPCR might for instance detect degradation products or viral RNA from non-productive dead cells.
- Fig. 1B: Could the authors please clarify what the data is relative to? Additionally, it would be helpful to specify whether the intervals represent SEM or SD.
- Fig 1I: The phenotypes may not be clear to non-specialists and should be described in more details.
- Is MSI1 expressed at birth in mice?
- Fig 2: These results show that mortality continues until day 21. It would be helpful for the authors to include viral load data at later time points to provide a better viral load kinetic profile. Since Fig C shows higher replication of mutants between days 3-6, with similar differences from days 6-9, it would be interesting to know if the wt virus continues to replicate while the mutants would not. More detailed kinetics in the brain would help address these questions.
- Fig 3: There is no Material/Methods section for the experiments with human brain organoids. This must be included. The claim that organoids infected with WT ZIKV shrank and lost their structure by day 5 would be more convincing if supporting size measurements were provided. Additionally, an explanation for the omission of MD1 in the analysis would be beneficial. After Fig. 5, the reintroduction of MD1 and MD2 appears without sufficient context, and further clarification is required.
- Fig 5: Some info on mortality/weight measurements after the challenge should be provided. The ELISPOT CONA technique should be briefly explained. What are the PRNT50 values and IgG titers obtained in infections using the wild type parental virus? Without this comparison, it is difficult to understand the advantages of using the MBD mutants as vaccine strains, as it is expected that the WT (which is only slightly more replicative) would yield comparable results.
- Fig 6: Could the authors clarify what the viral loads in the urine and blood of WT ZIKV are after 56 days? It seems that these datapoints are absent from B and C. They should be included if available.
- Fig S1: Staining of MSI1 (and ideally with ZIKV) in brain sections should be provided to better assess expression and tissue localization. Only staining in testis, placenta and retina are shown.
- In the vertical transmission model, only the brain is analyzed from the fetus. Without analyzing other fetal tissues, the claim about transmissibility should be tuned down as only neurotropism could be affected by the mutations. Measuring viral load in fetal tissues other than the brain could provide further insight.
- Could a ZIKV MBD1-2 double mutant be relevant?

Referee #1 (Remarks for Author):

In this manuscript, the authors proposed a novel live-attenuated vaccine strategy by targeting the tertiary structure of ZIKV viral RNA that interacts with a specific host protein Musashi1 (MSI1). The authors designed and engineered two mutants, MBD1 and MBD2, for vaccine immunogenicity and safety assays. MBD ZIKV mutants exhibit restricted replication in MSI1-expressing tissues while maintaining robust replication in MSI1-deficient cells (e.g., Vero cells). A single immunization of MBD ZIKV induced strong humoral and cellular immunity, protecting mice and non-human primates (NHPs) against wild type ZIKV challenge. The introduced mutations remained intact after serial passages in vitro and in vivo. Overall, this present study provides a proof-of-concept validation for the use of viral RNA-protein interactions as a rational attenuation strategy, and similar strategies may be applicable to other viruses.

The experiments are well designed, and the manuscript is clearly written. Nevertheless, there are few concerns that the authors need to address for the manuscript to warrant publication.

Response: We sincerely thank the reviewer for acknowledging the importance and novelty of our research and for the careful overview of our work.

1) The MSI1 expression patterns are the critical determinants for vaccine design. The major findings of the present manuscript were based on mouse model (partially validated in monkey), is there any difference in MSI1 expression in human or during development?

Response: Thanks for your insightful comments. The amino acid sequence identity of MSI1 between mouse and human is 99.45% (See below. The two RNA recognition domains (RRMs) are framed).

	1	10	20	30	40	50	60	70	80	90	
Mouse MSI1	METDAPQPGGLASPDSPHDPCKMFIGGLSWQTTQEGLEREYFGQFGEVKECLVMRDPLTKRSRGFGFVTFMDQAGVDKVLQSRHELDSKTIDPKVAFF										
Human MSI1	METDAPQPGGLASPDSPHDPCKMFIGGLSWQTTQEGLEREYFGQFGEVKECLVMRDPLTKRSRGFGFVTFMDQAGVDKVLQSRHELDSKTIDPKVAFF										
	RRM1										
	100	110	120	130	140	150	160	170	180	190	
Mouse MSI1	RRAQPKMVTTRKKIFVGGLSVNTTVEDVRYFEPQGVDDAMLMFDKTTNRHRGFGFVTFESEDIVEKVCEIHPHEINNKMVECKKAQPREVMSPTG										
Human MSI1	RRAQPKMVTTRKKIFVGGLSVNTTVEDVRYFEPQGVDDAMLMFDKTTNRHRGFGFVTFESEDIVEKVCEIHPHEINNKMVECKKAQPREVMSPTG										
	RRM2										
	200	210	220	230	240	250	260	270	280	290	
Mouse MSI1	SARGRSRVMPYGMDFMLGIGMLGYPGFQATTYASRSYTGGLAPGYTYQFPFRVERPLPSAPVLPETAIPLTAYGPMAAAAAAAVVVRTGSGHPW										
Human MSI1	SARGRSRVMPYGMDFMLGIGMLGYPGFQATTYASRSYTGGLAPGYTYQFPFRVERPLPSAPVLPETAIPLTAYGPMAAAAAAAVVVRTGSGHPW										
	300	310	320	330	340	350	360				
Mouse MSI1	TMAPPFGSTPFRITGGFLGTSFGPMAELYGAANQDSGVSSYISAASAPSTGFGHSLGGPLIATAFTNGYH										
Human MSI1	TMAPPFGSTPFRITGGFLGTSFGPMAELYGAANQDSGVSSYISAASAPSTGFGHSLGGPLIATAFTNGYH										

The organ distribution of *Msi1* mRNA is highly similar between mouse and human, as shown in previous studies^{1,2} and supported by our own results (**Fig. 1A and B**).

The developmental expression profile of *Msi1* mRNA has been characterized in the mouse brain¹. The mRNA level peaks at embryonic day 12 (E12) and subsequently declines gradually throughout development. In human embryonic brain, *MSI1* is abundant in neural precursors but absent from mature neurons³. We have mentioned these points in revision (**line 75-77**).

2) The authors proposed the tertiary structure of MBS2 mediates *MSI1* binding, and the *MBD2* mutant was generated by disrupting this tertiary structure. To strengthen this mechanistic claim, functional validation is required to demonstrate that *MSI1* binding to MBS2 is structurally dependent (i.e., conformation-sensitive) rather than solely sequence-dependent. This distinction is critical for establishing the specificity of the attenuation mechanism.

Response: Thanks for your constructive suggestion. We have clarified this point in our previous paper in *Nature Communication*, 2023⁴. We have generated two additional *MBD2* mutants, in which the AGAA loop was replaced with AGCU (P2-AGCU) or AGUU (P2-AGUU). These two mutants altered the MBS2 sequence while preserving its structure. *MSI1* binding analysis by isothermal titration calorimetry (ITC) demonstrated that both mutant RNAs retained strong binding affinity to *MSI1*, indicating that *MSI1* binding to MBS2 is structure-dependent but sequence-independent (**see below**).

3) Flavivirus xrRNAs (including MBS-containing elements) are known to perform multiple regulatory functions in viral replication. The authors are suggested to investigate whether MBD mutations affect other functions (e.g., sfRNA production) to rule out off-target effects.

Response: Thanks for your constructive suggestions. Accordingly, we profiled sfRNA expression of mutant versus WT viruses in human SH-SY5Y (MSI1-proficient) and A549 (MSI1-deficient) cell lines. Northern blot analysis demonstrated that identical sfRNA profiles between MBD mutant and WT viruses in both cellular contexts (see below and new Fig. EV3F, G). This results clearly demonstrate that the MBD mutations didn't infer the formation of sfRNAs. These new data were combined in revision as new Fig S3F and S3G.

Additionally, we have compared the viral replication kinetics of WT and MBD mutant ZIKV in A549 cells, which are MAVS-competent but deficient in MSI1, as well the induction of *IFN-β* and *IFIT1* mRNA by RT-qPCR at various time points post-infection. Both WT and MBD mutants exhibited similar replication kinetics and comparable

levels of *IFN-β* and *IFIT1* mRNA induction (see below and new Fig. EV3C-E). These results indicate that the attenuation phenotype is independent of host innate immunity. These new data were combined in revision as new Fig S3C to S3E.

4) While the manuscript highlights the broad applicability of this strategy, it would be beneficial for the authors to provide a prioritized list of specific viral candidates

Response: Thanks for the suggestion. Many pathogenic viruses contain RNA elements that bind host proteins. For example, Hepatitis C virus (HCV) UTRs and SARS-CoV-2 RNA genome bind to IGF2BP1 to promote viral translation. DENV2 3'UTR binds to TRIM25, G3BP1, G3BP2 and CAPRIN1 to avoid immune response. Targeting these viral RBP binding sites offers a promising platform or for LAVs design. We have discussed this in the Discussion section (line 302-308).

5) Figure.5: Did any adverse events (e.g., morbidity/mortality) occur during the immunization procedure?

Response: In agreement with the attenuation phenotype, all mice received MDB mutants or PBS immunization survived and no obvious adverse event was observed during the experimental period, while the WT group exhibited transient body weight reduction. We have amended this result in the revised manuscript (see below and the new Fig. 5B).

6) Figure.4G: were there any difference in the weight of fetal heads between the two groups?

Response: Thanks for the suggestion. Quantitative analysis showed that the head weight of WT group was significant lower than that of MBD2 group (see below).

Referee #2 (Comments on Novelty/Model System for Author):

This manuscript was enjoyable to read and is relevant considering that there are no available vaccines against Zika virus. The results are convincing. The attenuated strain (strikingly and interestingly) show no disease in mice although they replicate only relatively slightly less than the parental strain.

However, as detailed in my comments below, the proposed Musashi1-dependent mechanism behind this attenuation is not convincing. In particular, there is a possibility that the modification of the 3'UTR disrupts the capacity of the virus to interfere with the immune system (in addition to modulating replication efficiency), which would explain the phenotypes. This is properly not addressed here as well as in their previously published study). Several experiments are proposed in my remarks to the authors below.

The authors also need to better justify why these ZIKV mutants would be suitable attenuated vaccines as the parental strain is already not pathogenic and was excluded from the mice immunization/challenge study.

This is a nice study and addressing these concerns, would make this study suitable for publication EMBO Mol Med.

Response: Thanks for the encouraging comments and insightful suggestions.

Referee #2 (Remarks for Author):

In this manuscript, Chen and colleagues developed mutants of the Zika virus (ZIKV) by disrupting RNA motifs in the viral genome, which bind to the host protein Musashi-1 (MSI1). The same research group had previously shown that these RNA structures play a role in viral replication in MSI1-expressing cells. The authors generated the MBD1 mutant with a single mutation changing the motif, and MBD2 again with a single mutation in 3' UTR region of ZIKV RNA changing in that case the tertiary structure of a tetraloop. This study shows that the use of these attenuated mutants with structural changes to the Musashi-1 binding site (MBS) potentially offers an immune protection in both mice and monkey challenge infection models. Furthermore, they demonstrate that in tissues where ZIKV infection causes more severe pathology, such as in neural,

placental, and testicular tissues, these mutants replicate less than the parental strain, strikingly correlating with an abrogation of the pathogenesis. This allowed the authors to claim that the perturbation of MBS led to the absence of MSI1 binding in these MSI1-expressing tissues, which is necessary for proper viral replication. As a result, these mutants did not replicate efficiently, and the authors proposed that they could be safely administered as a vaccine.

The study highlights the significant potential of modifying the 3' UTR region of the viral genome to develop live attenuated virus vaccines. The data demonstrating tissue-specific responses to these modified sequences, driven by the ability of specific host proteins to bind these regions and modulate viral replication, further strengthens the appeal of the study concept. The *in vivo* phenotypes in terms of virus-induced symptoms are very convincing. However, some of the results especially regarding the mechanisms behind attenuation must be strengthened to make the conclusions more convincing. One concern is that the study's claim of tissue-specific viral attenuation is primarily based on RT-qPCR readouts, which may not always accurately reflect the actual efficacy of viral replication. Titration of infectious viral particles (using for instance TCID₅₀, focus-forming unit or plaque assays), at least for some key experiments (e.g., brain samples), would be highly beneficial in further supporting the attenuation phenotype of these mutants and help better assess the difference between WT and mutants ZIKV. The same applies for *in vitro* experiments. Since RT-qPCR also detects dead cells and not necessarily RNA active in replication (e.g. degradation products), viral titers (reflecting a productive viral replication cycle) might actually show in the end that attenuation is more pronounced than what is concluded here (only 10-fold with the RNA readout).

Response: Thanks for your suggestion. We have measured viral loads in the brain, testes, eyes and serum by standard plaque assay. The results showed that viral loads of the MBD mutants were significantly lower than those of the WT in brains, testes, and eyes, while the viral loads in the serum were comparable between the MBD mutants and WT (see below and new Fig. 1E-H). The plaque assay findings align with the viral RNA load data and further reinforce our original conclusions.

Additionally, we assessed the viral kinetics in A549, Y79 and Bewo cells using plaque assay (see below and new Fig. EV2B, EV3C and EV5B). All of these results strongly support our original conclusion regarding the attenuated phenotype of the MBD mutants.

The authors also assessed the impact of two ZIKV vaccine strains on one of the most severe ZIKV-associated complications, namely microcephaly, in a neonatal mouse model. They found that infection in the brain with low MSI1 expression, using the MD1 and MD2 strains, did not induce microcephaly, showing a 1-log reduction in viral RNA and the absence of the microcephaly phenotype. This complete restoration of survival and neonatal brain size with the mutants is very convincing. However, with only one Log₁₀ reduction in viral RNA levels, it is hard to comprehend how such relatively mild attenuation can lead to such drastic phenotypes. This raises doubts that this is only due to the replication capacity of these mutant viruses and further suggest that other mechanisms are at play. Given that neuroinflammation can drive developmental changes, a more detailed analysis would strengthen the claim that the

absence of microcephaly is due to reduced viral load rather than immune modulation. Additional data on the immune response, including interferons and key proinflammatory cytokines/chemokines with both wt and mutant viruses, would clarify whether the phenotypes associated with the mutants result from immune modulation or viral load reduction.

Response: Thanks for your constructive suggestion. We totally agree with the reviewer that host immune regulation may also contribute to the observed attenuation phenotype. Accordingly, we performed additional RNA-seq analysis to compare the transcriptional profiles of neonatal mouse brains infected with Mock, WT, or MBD2 ZIKV at 3 days post-infection. We first analyzed immune-related genes that were commonly upregulated across the WT and MBD2 infection groups compared to the Mock control. This analysis revealed that the magnitude of this upregulation was significantly attenuated in the MBD2 group relative to the WT group (**see below and new Appendix Fig. S2C, D**). These results indicate that MBD2 infection induced a significantly weaker immune response than WT infection.

In addition, several brain development-associated genes that were downregulated following WT infection were restored in the MBD2 infection group (**see below and new Appendix Fig. S2E**). Whether these phenotypes result from reduced viral RNA levels of the MBD mutant or from the 3'UTR modification requires further investigation. Nevertheless, these findings provide additional mechanistic explanation for the attenuated phenotype observed in MBD mutants.

The discrepancy between the differences in viral loads and phenotypes in vivo raises a fundamental question about UTR modifications and their potential impact on sRNA status, which can influence the immune response as well as virus-induced cell death. First, examining sRNA status in human MSI1-negative cells could further support their hypothesis like in their previous work with northern blots in hamster BHK-21

cells showing no impact on sfRNA expression profile. However, this must be done in immune-competent human cells (ideally in both MSI1-expressing and MSI1 deficient cells). This would help demonstrate that the viral attenuation of these mutants is not dependent on sfRNA expression. In the same line of idea, the authors show replication in MSI1-deficient Vero cells (Fig. S3) and use these data to claim that the attenuation is MSI1-dependent. This is not ideal because these cells are not human, do not have a functional innate immune system and are extremely efficient at producing viruses. Here, only the 72 hpi time point at which production is most probably saturated. To further support their claim, the authors must perform viral replication kinetics in human cells lacking MSI1 but still competent in MAVS-dependent innate immunity which would provide stronger evidence that attenuation is MSI1-dependent. Additionally, assessing immune response (gene or protein) in this context would strengthen the overall findings.

Response: Thanks for your constructive suggestions. Accordingly, we have profiled sfRNA expression of mutant versus WT viruses in human SH-SY5Y (MSI1-proficient) and A549 (MSI1-deficient) cell lines. Northern blot analysis showed identical sfRNA expression between the MBD mutants and WT viruses in both cellular contexts (see below and new Fig. EV3F, G).

Additionally, we compared the viral replication kinetics of WT and MBD mutant ZIKV in A549 cells, which are MAVS-competent but deficient in MSI1. We also measured the induction of *IFN- β* and its downstream gene *IFIT1* by RT-qPCR at various time points post-infection. Both WT and MBD mutants exhibited similar replication kinetics

and induced comparable levels of *IFN-β* and *IFIT1* (see below and new Fig. EV3C-E). These results indicate that the attenuation phenotype is dependent on MSI1 expression and independent of innate immunity.

This study suggests that the ZIKV mutant's vertical transmission capability is restricted. The authors first analyze the expression levels of MSI1 in the placenta and hypothesize that the presence of MSI1 in placenta may play a role in vertical transmission. In conclusion, they observe lower viral loads in the fetal heads and placenta of pregnant mice infected with MSD2 strain compared to the WT virus and base their claim of reduced transmission on this observation. The author only analyzed the maternal spleen and refer to results with A129 mice, which lack immunity compared to CD1 animals. To firmly conclude that vertical transmission is affected, further investigation across multiple organs is needed, particularly with those where no significant differences between with-type and mutant strains are observed. This would help clarify that the reduced viral load in the placenta and fetal heads of CD1 mice infected with MBD2 is not simply due to a lower amount of virus in the maternal periphery. The authors might consider measuring viral loads in the serum/blood to provide valuable insight as to whether there is generally less circulating virus in peripheral organs at this stage. Lastly, it would be helpful if the authors could clarify why MSD2 was prioritized for the vertical transmission study as the MSD1 mutant was omitted despite resulting in similar results in other experiments.

Response: Thanks for your insightful comments and suggestions. Accordingly, we have added measurements of viral RNA loads in maternal sera. The results showed that the viral RNA levels were similar between WT- and MBD2-infected maternal sera (see below and new Fig. 4D), suggesting that the reduced viral RNA loads observed in the

placentas and fetal heads are not due to lower levels of circulating virus in the maternal periphery.

The explanation for the omission of MBD1 has been incorporated into our revised manuscript (**line154-157**) and is shown below.

Given that MBD1 and MBD2 showed similar attenuation phenotypes in the aforementioned experiments, we selected the MBD2 mutant, which contains a disrupted tertiary structure of MBS2, for subsequent experiments to highlight the structural novelty and avoid redundancy.

The authors performed the entire study using the pre-epidemic (and presumably non-pathogenic) FSS13025 strain (Cambodia 2010). Did the authors consider testing more recent, microcephaly-associated strains in their models, such as H/PF/2013 or Brazilian strains? It would be highly relevant to assess whether prior immunization with FSS13025 MSD2 potentially offers protection against these more aggressive strains. Importantly, in Figure 5, the parental wt strain was not included in the analysis. In that context, it is really hard to judge what is the advantage of considering the mutants as vaccine strains compared to the wt since the latter replicates only a bit better (approximately 1 Log₁₀ as assessed by RT-qPCR) and is presumably not pathogenic in Human.

Response: Thanks for your helpful suggestion. We have conducted additional immunization and challenge experiment with a more recent American isolate in mice. The A129 mice were immunized with WT or MBD ZIKVs and subsequently challenged with the epidemic ZIKV strain VEN/2016 (an American isolate obtained from a patient returning from Venezuela in 2016). Changes in body weight and antibody responses were measured in mice immunized with the MBD mutants and

parental WT strain. All mice survived for 28 days post-immunization; and weight loss was only observed in mice immunized with the WT strain as expected (see below and new Fig. 5B). Importantly, the MBD mutants elicited antibody levels comparable to those induced by the WT strain (see below and new Fig. 5C and D). Following the challenge, all immunized mice survived without detectable viremia or weight loss, whereas the mock-immunized group exhibited significant viremia and weight loss and eventually succumbed (see below and new Fig. 5E - 5G). Together with the in vitro and in vivo viral replication data from Fig 1-4, these results demonstrate that the MBD mutants exhibit marked attenuation while maintaining robust immunogenicity.

Other comments:

-The authors regularly use the word "titer" when measuring the viral RNA load (e.g. line 133). This is not correct since it does not necessarily correlate with the abundance of infectious particles and a productive and complete ZIKV life cycle. RT-qPCR might for instance detect degradation products or viral RNA from non-productive dead cells.

Response: Thanks for the reminder, we have corrected this error throughout the manuscript.

- Fig. 1B: Could the authors please clarify what the data is relative to? Additionally, it would be helpful to specify whether the intervals represent SEM or SD.

Response: The data are relative to the geometric average of two reference genes, B2m and GAPDH. The intervals represent SD. These clarifications have been incorporated into the Material section (**line 424-425**) and the figure legend (**Line 886**) in the revised manuscript.

- Fig 1I: The phenotypes may not be clear to non-specialists and should be described in more details.

Response: We have provided a more comprehensive description of the results presented of Fig 1I (**line132-136**) (and see below).

“In the testicular tissues infected with WT ZIKV, substantial ZIKV E-positive cells were observed in adjacent clusters, with signals distributed throughout the seminiferous epithelium. In contrast, only a small number of positive cells were detected in a scattered distribution in the testicular tissues infected with MBD1 or MBD2 (Fig. 1I)”

- Is MSI1 expressed at birth in mice?

Response: Yes, MSI1 is expressed at birth in mice as demonstrated in several studies^{1,5,6}.

- Fig 2: These results show that mortality continues until day 21. It would be helpful for the authors to include viral load data at later time points to provide a better viral load kinetic profile. Since Fig C shows higher replication of mutants between days 3-6, with similar differences from days 6-9, it would be interesting to know if the wt virus continues to replicate while the mutants would not. More detailed kinetics in the brain would help address these questions.

Response: Thanks for your suggestions. We have detected the viral RNA loads on day 12 post infection in the original study. The results showed that the viral RNA loads of WT and MBD mutants declined to similar levels at this time point. These data have been incorporated into the revised manuscript as part of the **new Fig. 2C (see below)**.

- Fig 3: There is no Material/Methods section for the experiments with human brain organoids. This must be included. The claim that organoids infected with WT ZIKV shrank and lost their structure by day 5 would be more convincing if supporting size measurements were provided. Additionally, an explanation for the omission of MD1 is the analysis would be beneficial. After Fig. 5, the reintroduction of MD1 and MD2 appears without sufficient context, and further clarification is required.

Response: Sorry for missing this information. The experimental method for brain organoids has been provided in the revised manuscript (line 430-442). Size measurements are also included, and the results strongly support our conclusion (see below and new Fig. 3B).

The explanation for the omission of MBD1 has been incorporated into our revised manuscript (line 154-157) and is shown below.

Given that MBD1 and MBD2 showed similar attenuation phenotypes in the aforementioned experiments, we selected the MBD2 mutant, which contains a disrupted tertiary structure of MBS2, for subsequent experiments to highlight the structural novelty and avoid redundancy.

The explanation for the reintroduction of MD1 and MD2 in Fig. 5 has been incorporated into our revised manuscript (**line221-223**) and is shown below.

Both MBD1 and MBD2 were analyzed in the vaccine immunization study to validate that structure-targeted attenuation (MBD2) confers protection comparable to conventional sequence disruption (MBD1).”

- Fig 5: Some info on mortality/weight measurements after the challenge should be provided. The ELISPOT CONA technique should be briefly explained. What are the PRNT50 values and IgG titers obtained in infections using the wild type parental virus? Without this comparison, it is difficult to understand the advantages of using the MBD mutants as vaccine strains, as it is expected that the WT (which is only slightly more replicative) would yield comparable results.

Response: Thanks for the suggestion. We have performed additional animal experiment to profile the mortality/weight changes. Accordingly, A129 mice were immunized with WT and MBD ZIKV strains and subsequently challenged with the epidemic ZIKV strain VEN/2016 (an American isolate from a patient returning from Venezuela in 2016). Mortality and body weight changes following the challenge were assessed, and the results have been included in the revised manuscript (see **below and new Fig. 5F and G**).

Concanavalin A (Con A) was used as a positive control in the ELISPOT assay because it binds to T cell surface glycoproteins, leading to T cell activation and cytokine production. This has been explained in the material and methods (**line 473-474**).

- Fig 6: Could the authors clarify what the viral loads in the urine and blood of WT ZIKV are after 56 days? It seems that these datapoints are absent from B and C. They should be included if available.

Response: Thanks for your suggestion, and we have included these data in the revised Fig. 6B and C (see below).

- Fig S1: Staining of MSI1 (and ideally with ZIKV) in brain sections should be provided to better assess expression and tissue localization. Only staining in testis, placenta and retina are shown.

Response: We have provided MSI1 staining of brain sections in the revised Fig. EV1 (see below)

- In the vertical transmission model, only the brain is analyzed from the fetus. Without analyzing other fetal tissues, the claim about transmissibility should be tuned down as

only neurotropism could be affected by the mutations. Measuring viral load in fetal tissues other than the brain could provide further insight.

Response: Thank you for your comment. We agree with the reviewer that further analysis with other fetal tissues would be helpful. However, at embryonic day 13.5 (E13.5), the organ systems in mouse embryos are still in a rudimentary stage, making it technically challenging to precisely dissect individual organ for quantitative virological assessment. Accordingly, we have moderated our conclusion regarding transmissibility to: “These results clearly demonstrate highly restricted infection of neural tissues during vertical transmission of MBD ZIKV”

- Could a ZIKV MBD1-2 double mutant be relevant?

Response: Thank you for your suggestion. We did generate a double mutant ZIKV combining MBD1 and MBD2, referred to MBD12. The MBD12 can be successfully recovered and grows well in Vero cells. Given that the single MBD1 and MBD2 mutants exhibited comparable attenuation phenotypes and further characterization of the double mutant is unlikely to yield additional mechanistic insights, we refrained from conducting redundant experiments. Such experiment may be involved in future studies.

References

- 1 Sakakibara, S. *et al.* Mouse-Musashi-1, a neural RNA-binding protein highly enriched in the mammalian CNS stem cell. *Dev Biol* **176**, 230-242, doi:10.1006/dbio.1996.0130 (1996).
- 2 Good, P. *et al.* The human Musashi homolog 1 (MSI1) gene encoding the homologue of Musashi/Nrp-1, a neural RNA-binding protein putatively expressed in CNS stem cells and neural progenitor cells. *Genomics* **52**, 382-384, doi:10.1006/geno.1998.5456 (1998).
- 3 Chavali, P. L. *et al.* Neurodevelopmental protein Musashi-1 interacts with the Zika genome and promotes viral replication. *Science* **357**, 83-88, doi:10.1126/science.aam9243 (2017).

- 4 Chen, X. *et al.* Zika virus RNA structure controls its unique neurotropism by bipartite binding to Musashi-1. *Nat Commun* **14**, 1134, doi:10.1038/s41467-023-36838-w (2023).
- 5 Sakakibara, S. & Okano, H. Expression of neural RNA-binding proteins in the postnatal CNS: implications of their roles in neuronal and glial cell development. *J Neurosci* **17**, 8300-8312, doi:10.1523/JNEUROSCI.17-21-08300.1997 (1997).
- 6 Kaneko, Y. *et al.* Musashi1: an evolutionally conserved marker for CNS progenitor cells including neural stem cells. *Dev Neurosci* **22**, 139-153, doi:10.1159/000017435 (2000).

15th Aug 2025

Dear Prof. Qin,

Thank you for the submission of your revised manuscript to EMBO Molecular Medicine and please accept my apologies for the delay in getting back to you due to the holiday season. I am pleased to inform you that we will be able to accept your manuscript pending the following final amendments:

1) Figures:

- Please remove all figures from the main manuscript file and only leave their legends and the end of the file.
- During our routine image checks, we noticed that the microscopy panels across the figure set appear pixelated. This is a common result of converting original 16-bit TIFF images to RGB format for publication, and while not a cause for concern, it can sometimes give the impression of image alteration to critical readers. To avoid any misunderstanding and to meet EMBO Press standards, we kindly ask that you resubmit the complete figure set at the captured original data resolution.

2) In the main manuscript file, please do the following:

- Please address all comments suggested by our data editors listed below:

o Figure legends:

1. Please note that the exact p values are not provided in the legends of figures 4G, H; 5G, H.

2. Please indicate the statistical test used for data analysis in the legends of figures S2A, B.

3. Please note that n=2 in figure 1A.

4. Please note that the error bars are not defined in the legends of figure 1A.

- Add up to 5 keywords.

- Add callouts for the Figure 5H.

- Please correct the order and headings of the manuscript sections to: Abstract, Introduction, Results, Discussion, Methods, Acknowledgements, Disclosure and competing interests statement, References, Figure legends, Tables and their legends, Expanded View Figure legends.

- Rename "Conflict of interests" to "Disclosure and competing interests statement". We updated our journal's competing interests policy in January 2022 and request authors to consider both actual and perceived competing interests. Please review the policy <https://www.embopress.org/competing-interests> and update your competing interests if necessary.

- Author contributions: Please remove it from the manuscript and specify author contributions in our submission system. CRediT has replaced the traditional author contributions section because it offers a systematic machine-readable author contributions format that allows for more effective research assessment. You are encouraged to use the free text boxes beneath each contributing author's name to add specific details on the author's contribution. More information is available in our guide to authors:

<https://www.embopress.org/page/journal/17574684/authorguide#authorshipguidelines>

- Indicate in legends exact n and exact p values, not a range, along with the statistical test used. To keep the figures "clear" some authors found providing an Appendix table Sx with all exact p-values preferable. You are welcome to do this if you want to.

- Please use the following format to report the accession number of your data:

[data type]: [full name of the resource] [accession number/identifier] [(doi or URL or identifiers.org/DATABASE:ACCESSION)]

Please check "Author Guidelines" for more information.

<https://www.embopress.org/page/journal/17574684/authorguide#availabilityofpublishedmaterial>

3) Funding: Please make sure that information about all sources of funding are complete in both our submission system and in the manuscript. 2018YFA0900801 is missing in the Acknowledgements and 2023YFC2305900 is missing in our submission system. Please correct.

4) Reagent Table: Remove it from the main manuscript file and only leave separately uploaded file.

5) Appendix: Please merge the appendix figure files into one PDF file, add the legends under each figure and add a table of contents with page numbers. Remove the appendix figure legends from the main manuscript text.

6) The Paper Explained: Please add it to the main manuscript file.

7) Synopsis:

- Please make the synopsis image symmetrical by adding the white background on the right side to match the left side of the image.

8) As part of the EMBO Publications transparent editorial process initiative (see our Editorial at

<http://embomolmed.embopress.org/content/2/9/329>), EMBO Molecular Medicine will publish online a Review Process File (RPF) to accompany accepted manuscripts. This file will be published in conjunction with your paper and will include the anonymous referee reports, your point-by-point response and all pertinent correspondence relating to the manuscript. Let us know whether you agree with the publication of the RPF and as here, if you want to remove or not any figures from it prior to publication.

9) Please provide a point-by-point letter INCLUDING my comments as well as the reviewer's reports and your detailed responses (as Word file).

I look forward to reading a new revised version of your manuscript as soon as possible.

Yours sincerely,

Zeljko Durdevic

Zeljko Durdevic
Senior Editor
EMBO Molecular Medicine

*** Instructions to submit your revised manuscript ***

- 1) a .docx formatted version of the manuscript text (including Figure legends and tables)
- 2) Separate figure files*
- 3) supplemental information as Expanded View and/or Appendix. Please carefully check the authors guidelines for formatting Expanded view and Appendix figures and tables at <https://www.embopress.org/page/journal/17574684/authorguide#expandedview>
- 4) a letter INCLUDING the reviewer's reports and your detailed responses to their comments (as Word file).
- 5) The paper explained: EMBO Molecular Medicine articles are accompanied by a summary of the articles to emphasize the major findings in the paper and their medical implications for the non-specialist reader. Please provide a draft summary of your article highlighting
 - the medical issue you are addressing,
 - the results obtained and
 - their clinical impact.This may be edited to ensure that readers understand the significance and context of the research. Please refer to any of our published articles for an example.
- 6) Author contributions: the contribution of every author must be detailed in a separate section.
- 7) EMBO Molecular Medicine now requires a complete author checklist (<https://www.embopress.org/page/journal/17574684/authorguide>) to be submitted with all revised manuscripts. Please use the checklist as guideline for the sort of information we need WITHIN the manuscript. The checklist should only be filled with page numbers where the information can be found. This is particularly important for animal reporting, antibody dilutions (missing) and exact values and n that should be indicated instead of a range.
- 8) Every published paper now includes a 'Synopsis' to further enhance discoverability. Synopses are displayed on the journal

webpage and are freely accessible to all readers. They include a short stand first (maximum of 300 characters, including space) as well as 2-5 one sentence bullet points that summarise the paper. Please write the bullet points to summarise the key NEW findings. They should be designed to be complementary to the abstract - i.e. not repeat the same text. We encourage inclusion of key acronyms and quantitative information (maximum of 30 words / bullet point). Please use the passive voice. Please attach these in a separate file or send them by email, we will incorporate them accordingly.

You are also welcome to suggest a striking image or visual abstract to illustrate your article. If you do please provide a jpeg file 550 px-wide x 300-600px high.

9) A Conflict of Interest statement should be provided in the main text

10) Please note that we now mandate that all corresponding authors list an ORCID digital identifier. This takes <90 seconds to complete. We encourage all authors to supply an ORCID identifier, which will be linked to their name for unambiguous name identification.

Currently, our records indicate that the ORCID for your account is 0000-0002-0632-2807.

Link Not Available

11) Include a Reagents and Tools Table as part of the Methods section, which can be downloaded from our author guidelines (<https://www.embopress.org/page/journal/17574684/authorguide#structuredmethods>)

Photos 400-800 DPI

*Additional important information regarding figures and illustrations can be found at

<https://bit.ly/EMBOPressFigurePreparationGuideline>. See also figure legend preparation guidelines:

<https://www.embopress.org/page/journal/17574684/authorguide#figureformat>

***** Reviewer's comments *****

Referee #1 (Remarks for Author):

Dr. Cheng-Feng Qin and colleagues have fully addressed all reviewers' comments and incorporated the requested revisions into the manuscript. These changes have significantly improved the manuscript, now meeting the high standards of EMBO Molecular Medicine. I therefore recommend its immediate acceptance for publication in the journal.

Referee #2 (Comments on Novelty/Model System for Author):

The study is well technically performed and uses state-of-the art mouse development and monkey in vivo models. It is highly relevant since identifying a novel vaccine Zika virus strain that is less pathogenic in the fetal brain. I chose medium for novelty as it deals with previously reported mutants and concepts. But this does not minimize the suitability of this manuscript for publication in EMBO Mol Med

Referee #2 (Remarks for Author):

The authors have answered all my comments appropriately with very convincing new data. Overall, this is a very nice study.

The authors addressed the remaining editorial issues.

20th Aug 2025

Dear Prof. Qin,

We are pleased to inform you that your manuscript is accepted for publication and is now being sent to our publisher to be included in the next available issue of EMBO Molecular Medicine.

Zeljko Durdevic
Senior Editor
EMBO Molecular Medicine
